# OCEAN: Online Multi-modal Root Cause Analysis for Microservice Systems

## Abstract

Root Cause Analysis (RCA) is essential for pinpointing the root causes of failures in microservice systems. Traditional data-driven RCA methods are typically limited to offline applications due to high computational demands, and existing online RCA methods handle only single-modal data, overlooking complex interactions in multi-modal systems. In this paper, we introduce OCEAN, a novel online multi-modal causal structure learning method for root cause localization. OCEAN employs a dilated convolutional neural network to capture long-term temporal dependencies and graph neural networks to learn causal relationships among system entities and key performance indicators. We further design a multi-factor attention mechanism to analyze and reassess the relationships among different metrics and log indicators/attributes for enhanced online causal graph learning. Additionally, a contrastive mutual information maximization-based graph fusion module is developed to effectively model the relationships across various modalities. Extensive experiments on three real-world datasets demonstrate the effectiveness and efficiency of our proposed method.

## 1 Introduction

Root Cause Analysis (RCA) is crucial for identifying the underlying causes of system failures and ensuring the high performance of microservice systems (Wang et al., 2023a; Li et al., 2021; Wang et al., 2023c). Traditional manual root cause analysis is labor-intensive, costly, and error-prone, given the complexity of microservice systems and the extensive volume of data involved. Consequently, effective and efficient root cause analysis methods are vital for pinpointing failures in complex microservice systems and mitigating potential financial losses when system faults occur.

Previous studies in data-driven RCA, particularly those utilizing causal discovery techniques, have primarily focused on constructing causal or dependency graphs (Ikram et al., 2022; Lu et al., 2017; Li et al., 2021; Soldani & Brogi, 2022; Wang et al., 2023c; Zheng et al., 2024a). These graphs depict the causal links between different system entities and key performance indicators (KPIs), thereby enabling the tracing of underlying causes through these structures. For instance, Wang *et al.* (Wang et al., 2023c) developed a hierarchical graph neural network method that automatically identifies causal relationships both within and between networks to help pinpoint root causes.

Despite significant advances, most of these approaches are designed for offline use and face challenges with real-time implementation in microservice systems due to high computational demands. To address this, Wang *et al.* (Wang et al., 2023a) introduced an online RCA method that decouples state-invariant and state-dependent information and incrementally updates the causal graph. Li *et al.* (Li et al., 2022) developed a causal Bayesian network that leverages system architecture knowledge to mitigate potential biases toward new data. However, these online RCA methods are limited to handling single-modal data.

Recently, multi-modal data, such as system metrics and logs, are commonly collected from microservice systems, revealing the complex nature of system failures (Zheng et al., 2024a). For instance, failures such as "Database Query Failures" might be overlooked if only system metrics are considered, whereas issues like "Disk Space Full" are more effectively identified through combined analysis of metrics and logs. This underscores the importance of using multi-modal data for a thorough understanding of system failures. By integrating information from various sources, we can detect the abnormal patterns of system failures that might not be evident when analyzing single-modal data.

To bridge this gap, this paper aims to propose an online multi-modal causal structure learning method for identifying root causes in microservice systems. Formally, given the system KPI data along with multi-modal microservice data including metrics and log data, our objective is to develop an online multi-modal causal graph that identifies the top $k$ system entities most relevant to the system KPI. Three major challenges exist in this task. (C1) **Capturing Long-term Temporal Dependencies**: Current auto-regressive based RCA methods (Wang et al., 2023a; Zheng et al., 2024a) are limited to capturing short-term temporal dependencies. However, some system faults, such as Distributed Denial of Service (DDoS) attacks, may persist for extended periods. Effectively capturing these long-term temporal dependencies is crucial for identifying various types of system faults. (C2) **Capturing the Correlation of Multi-dimensional Factors**: Existing RCA approaches (Ikram et al., 2022; Wang et al., 2023c; Zheng et al., 2024a) often analyze abnormal patterns from multiple factors individually, such as CPU usage or memory usage from system metrics and frequency or golden signal from system logs, overlooking potential relationships among these factors from both modalities. Furthermore, these methods often consider all factors as equally important; however, in real applications, certain factors prove to be considerably more crucial than others. It is vital, therefore, to reassess the contributions of each factor to the learning of causal structures. (C3) **Learning Multi-modal Causal Structures**: Effectively capturing the relationships between different modalities in an online setting is crucial. Simply combining causal graphs from individual modalities can be problematic, especially if one modality is of lower quality.

To tackle these challenges, we introduce OCEAN, Online Multi-modal Causal Structure LEArNing, for root cause identification in microservice systems. Specifically, we propose to encode long-term temporal dependencies using a dilated convolutional neural network (Yu & Koltun, 2016) and forecast future values based on the $p$-th lagged historical data. We further develop a multi-factor attention mechanism to analyze the correlations among various factors and reassess their importance for causal graph learning. Additionally, we propose a contrastive mutual information estimation technique to model the relationships of different modalities. Our contributions can be summarized as follows:

- We introduce a novel online framework for multi-modality root cause analysis.
- We propose employing a dilated convolutional neural network to capture long-term temporal dependencies and graph neural networks to model causal relations among system entities.
- We design a multi-factor attention mechanism to analyze the relationships among different factors and reassess their impact on online causal graph learning.
- We develop graph fusion techniques with contrastive multi-modal learning to model the relationships between different modalities and assess their importance.
- Extensive experiments on three real-world datasets demonstrate the effectiveness and efficiency of our proposed method.

## 2 PRELIMINARY AND RELATED WORK

**Key Performance Indicator (KPI)**. In a microservice system, KPIs serve as invaluable metrics for assessing the effectiveness and productivity of the architecture (Podgórski, 2015). They play an indispensable role in monitoring and managing different aspects of microservices to uphold optimal performance levels. Common KPIs encompass latency and service response time. High values in these metrics typically indicate suboptimal system performance or potential system failure.

**Entity Metrics**. Entity metrics are the measurable time-series attributes that provide insights into the performance and status of services within a system (Bogner et al., 2017). These entities encompass various components such as physical machines, containers, virtual machines, and pods. In microservice architectures, typical entity metrics include CPU utilization, memory usage, disk I/O activity, packet transmission rate, and etc. These metrics are extensively employed to detect anomalous behavior and pinpoint potential causes of system failures in microservice environments (Wang et al., 2023a;b; Zheng et al., 2024a; Soldani & Brogi, 2023; Liu et al., 2021).

**Root Cause Analysis**. Current root cause analysis (RCA) methods can be categorized into two main branches: single-modal RCA methods and multi-modal RCA methods. Single-modal RCA methods primarily investigate causal relationships among system components using one type of data only (Sporleder et al., 2019; Duan et al., 2020; Meng et al., 2020; Soldani & Brogi, 2022; Aggarwal

Table 1: Notation Table

| | |
|---|---|
| $\boldsymbol{X}_M^0$ | the historical metric data |
| $\boldsymbol{X}_M^i$ | the $i$-th batch of the system metric |
| $\boldsymbol{X}_L^0$ | the historical log data |
| $\boldsymbol{X}_L^i$ | the $i$-th batch of system log |
| $T_1$ | the length of the historical metric time-series data |
| $T_2$ | the length of the batch for the system metric |
| $n - 1$ | the number of system entities |
| $\mathcal{T}$ | the total number of batches |
| $d_M$ | the number of different system metric features |
| $d_L$ | the number of different system log features |
| $\mathbf{y}$ | the system Key Performance Indicator |
| $\mathcal{G} = \{\mathcal{V}, \mathcal{A}\}$ | the causal graph |
| $\mathcal{A}$ | the adjacency matrix in the causal graph |

et al., 2020; Li et al., 2021). For instance, Liu *et al.* (Liu et al., 2021) generate a service call graph based on domain-specific software and rules, while Wang *et al.* (Wang et al., 2023c) construct causal networks from time series data. However, these methods often exhibit suboptimal performance due to their reliance on single-modal data. To enhance accuracy, recent research integrates multi-modal data for RCA (Yu et al., 2023; Hou et al., 2021; Zheng et al., 2024a). Nezha (Yu et al., 2023) and PDiagnose (Hou et al., 2021) extract and combine information from each modality individually, while MULAN (Zheng et al., 2024a) and MM-DAG (Lan et al., 2023) consider interactions among modalities, constructing comprehensive causal graphs. Despite notable progress, these approaches are implemented offline, necessitating extensive data collection and retraining for new faults. Wang *et al.* (Wang et al., 2023a) enable online root cause identification by decoupling state-invariant and state-dependent information to learn a causal graph for root cause identification. However, their focus remains on single-modal data. Recently, large language model (LLM)-based approaches have emerged as a new research direction for learning causal relations in root cause identification, owing to the success of LLMs in tackling complex tasks (Chen et al., 2024; Shan et al., 2024; Goel et al., 2024; Zhou et al., 2024; Roy et al., 2024; Wang et al., 2024). For example, Chen *et al.* (Chen et al., 2024) introduce RCACopilot, an on-call system powered by LLMs to automate RCA for cloud incidents. Similarly, Shan *et al.* (Shan et al., 2024) propose an approach that first identifies log messages indicating configuration-related errors, then localizes suspected root-cause configuration properties based on these log messages and LLM-generated configuration settings. While LLMs could effectively learn temporal dependencies within the metric data of individual system entities, they often struggle to capture interdependencies—such as causal relationships between different system entities—leading to higher computational costs. Unlike existing RCA methods, this paper addresses the online multi-modal RCA problem by uniquely modeling long-term temporal dependencies while simultaneously capturing the cross-modal correlation of multiple factors.

## 3 METHODOLOGY

In this section, we first present the problem statement and then introduce OCEAN, an online causal structural learning method designed to identify root causes using multi-modal data. We propose three modules to tackle the challenges outlined in the introduction: long-term temporal causal structure learning, multi-factor attention mechanism, and contrastive multi-modal learning. Subsequently, we identify potential root causes through the network propagation-based root cause identification module and establish stopping criteria. The overview of the proposed OCEAN is provided in Figure 1.

### 3.1 PROBLEM STATEMENT

Let $\mathcal{X}_M = \{\boldsymbol{X}_M^0, \boldsymbol{X}_M^1, ..., \boldsymbol{X}_M^{\mathcal{T}}\}$ represent $\mathcal{T} + 1$ multi-variate time series data for entity metrics. Here, $\boldsymbol{X}_M^0$ is the historical metric data, and $\boldsymbol{X}_M^i$, $i \in [1, \ldots, \mathcal{T}]$, is the $i^{\text{th}}$ batch for the metric data, with $T_1$ denoting the length of historical metric data, $T_2$ the length of each batch, $n - 1$ the number of system entities, and $d_M$ the number of different system metric features. Similarly, $\mathcal{X}_L = \{\boldsymbol{X}_L^0, \boldsymbol{X}_L^1, ..., \boldsymbol{X}_L^{\mathcal{T}}\}$ represents $\mathcal{T} + 1$ multi-variate time series data for system logs. Assuming preprocessing has converted the logs into multi-variate time series data (details of converting log data into time series can be found in Appendix C), $\boldsymbol{X}_L^0$ is the historical log data, and $\boldsymbol{X}_L^i$, $i \in [1, \ldots, \mathcal{T}]$, is the $i^{\text{th}}$ batch for system logs, where $d_L$ is the number of different log attributes/features. Notice

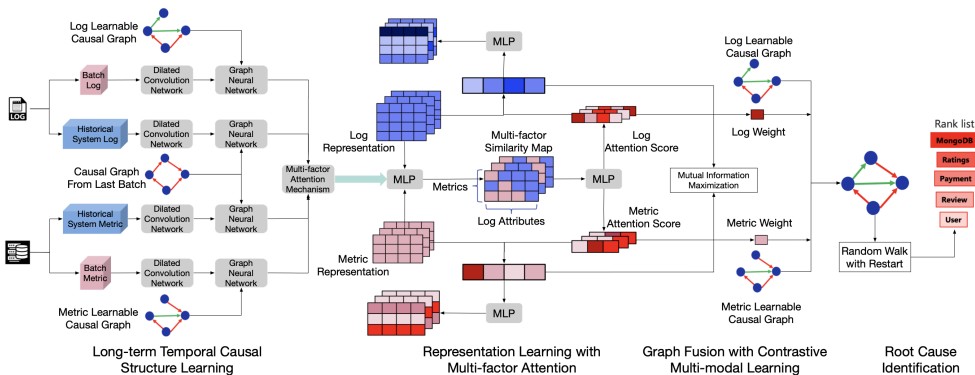

Figure 1: The overview of the proposed framework OCEAN with four main modules: long-term temporal causal structure learning, representation learning with multi-factor attention, graph fusion with contrastive multi-modal learning, and network propagation-based root cause localization.

that in the offline setting, only the historical data is available for model training, but in the online setting, the batch data is also available for the root cause analysis. The system KPI is denoted as $\mathbf{y} = \{\boldsymbol{y}^0, \boldsymbol{y}^1, ..., \boldsymbol{y}^{\mathcal{T}}\}$, with $\boldsymbol{y}^0$ and $\boldsymbol{y}^i$, $i \in [1, ..., \mathcal{T}]$, representing KPI data with lengths $T_1$ and $T_2$, respectively. Our goal is to construct a causal graph $\mathcal{G} = \{\mathcal{V}, \mathcal{A}\}$ to identify the top $k$ system entities most relevant to $\mathbf{y}$. Here, $\mathcal{V}$ represents the set of vertices, $\mathcal{A} \in \mathbb{R}^{n \times n}$ denotes the adjacency matrix, and $n$ is the total number of entities plus the system KPI. To simplify, we replicate the KPI $d_M$ times to match the number of metrics. This allows us to concatenate the system metric time-series data and KPI, yielding $\hat{\boldsymbol{X}}_M^0 \in \mathbb{R}^{n \times d_M \times T_1}$ and $\hat{\boldsymbol{X}}_M^i \in \mathbb{R}^{n \times d_M \times T_2}$ for metric data. Similarly, we combine the system log time-series data and KPI, denoted as $\hat{\boldsymbol{X}}_L^0 \in \mathbb{R}^{n \times d_L \times T_1}$ and $\hat{\boldsymbol{X}}_L^i \in \mathbb{R}^{n \times d_L \times T_2}$. Online RCA is a crucial step in system diagnosis post online fault/failure detection, which falls beyond the scope of this work. Here, we employ the Multivariate Singular Spectrum Analysis (MSSA) model (Alanqary et al., 2021), a state-of-the-art method for online failure detection, to identify the triggers for the root cause analysis process. We summarize the notation in Table 1.

## 3.2 Long-term Temporal Causal Structure Learning

To capture temporal causal relations among various system entities and KPIs, the Vector Autoregression Model (VAR) (Stock & Watson, 2001) is often employed (Wang et al., 2023a;b; Zheng et al., 2024a) due to its effectiveness in capturing dynamic interactions among variables in time series data. Specifically, given the two-way matrix $\boldsymbol{X}_{M,i} \in \mathbb{R}^{n \times T_1}$ for the $i^{\text{th}}$ system metric, our objective is to minimize a VAR-based loss function as follows:

$$\hat{\boldsymbol{X}}_{M,i}^t = \boldsymbol{W}^1 \boldsymbol{X}_{M,i}^{t-1} + \boldsymbol{W}^2 \boldsymbol{X}_{M,i}^{t-2} + \cdots + \boldsymbol{W}^{t-1} \boldsymbol{X}_{M,i}^1 + \epsilon$$

$$\mathcal{L}_{var} = \sum_{i=1}^{d_M} ||\boldsymbol{X}_{M,i}^t - F(\hat{\boldsymbol{X}}_{M,i}^t, \mathcal{A}, \theta)||^2 \tag{1}$$

where $\hat{\boldsymbol{X}}_{M,i}^t$ represents the prediction for the $i^{\text{th}}$ system metric, $\boldsymbol{W}^i \in \mathbb{R}^{n \times n}$ denotes the weight matrix, $\epsilon \in \mathbb{R}^n$ signifies the error variable, $F(\cdot)$ represents the graph neural network (Kipf & Welling, 2017) parameterized by $\theta$, $\mathcal{A}$ is the learnable causal graph capturing the relationships among node entities and KPIs, and $\boldsymbol{X}_{M,i}^t$ denotes the future value.

This model is also known as the $t^{\text{th}}$ order VAR-based model, where $t$ defines the range of temporal dependencies it can capture. However, a recent study (Lin et al., 2020) indicates that as the time lag $t$ increases, the autoregressive model becomes computationally expensive, making it challenging to capture long-term temporal dependencies in online settings. Similarly, many existing temporal modeling techniques, such as Recurrent Neural Networks (RNNs) and Transformers (Vaswani et al., 2017), also incur high computational costs, limiting their applicability for online root cause analysis (see Appendix A for detailed discussion).

To address this issue, we propose a module based on **dilated convolution** and **graph neural networks** to efficiently capture the long-term temporal dependencies and causal relations among system entities

and KPIs. Different from the VAR-based methods (Wang et al., 2023a;b; Zheng et al., 2024a) that take as input the 2-way matrix, we propose to capture the long-term temporal dependency via 3-way tensors. Specifically, given four 3-way tensors (*i.e.*, historical metric data $\hat{\boldsymbol{X}}_M^0$, historical log data $\hat{\boldsymbol{X}}_L^0$, the current batch of metric data $\hat{\boldsymbol{X}}_M^i$ and log data $\hat{\boldsymbol{X}}_L^i$), we follow the idea of LSTM (Hochreiter & Schmidhuber, 1997) and Gated Temporal Convolutional Network (TCN) (Wu et al., 2019) to model the temporal dependency for the historical and current batches of time series for two modalities as follows:

$$g(\boldsymbol{x}, \boldsymbol{f}) = \boldsymbol{x} * \boldsymbol{f} = \sum_{\tau=0}^{K-1} \boldsymbol{f}(\tau) \cdot \boldsymbol{x}(t - d \times \tau) \tag{2}$$

$$\boldsymbol{H}_v^0 = \tanh(g(\hat{\boldsymbol{X}}_v^0, \boldsymbol{f}_1)) \odot \sigma_1(g(\hat{\boldsymbol{X}}_v^0, \boldsymbol{f}_2)) \tag{3}$$

$$\boldsymbol{H}_v^i = \tanh(g(\hat{\boldsymbol{X}}_v^i, \boldsymbol{f}_3)) \odot \sigma_1(g(\hat{\boldsymbol{X}}_v^i, \boldsymbol{f}_4)) \tag{4}$$

$$\hat{\boldsymbol{O}}_v^0 = MFL(\hat{\boldsymbol{H}}_v^0), \hat{\boldsymbol{O}}_v^i = MFL(\hat{\boldsymbol{H}}_v^i) \tag{5}$$

where $\boldsymbol{f} \in \mathbb{R}^K$ represents the 1-D kernel, $d$ is the dilation factor controlling the skipping distance, $\odot$ denotes the Hadamard product, $v \in \{M, L\}$, $\sigma(x) = \frac{1}{(1+e^{-x})}$ is the sigmoid function, and $\tanh(x) = \frac{e^x - e^{-x}}{e^x + e^{-x}}$ is the tanh function. $\boldsymbol{f}_1, \boldsymbol{f}_2, \boldsymbol{f}_3$, and $\boldsymbol{f}_4$ are 1-D kernels of the dilated convolution networks. $\boldsymbol{H}_v^0 \in \mathbb{R}^{n \times d_v \times T_3}$ and $\boldsymbol{H}_v^i \in \mathbb{R}^{n \times d_v \times T_4}$ represent the historical time series and the $i^{\text{th}}$ batch of streaming time series for the modality $v$, respectively. $T_3$ and $T_4$ are the output dimensions of the dilated convolution networks. Additionally, $MFL(\cdot)$ denotes the representation learning with multi-factor attention module, aiming to encode the correlation of different metrics into the representations $\hat{\boldsymbol{O}}_v^0 \in \mathbb{R}^{n \times d_v \cdot T_3}$ and $\hat{\boldsymbol{O}}_v^i \in \mathbb{R}^{n \times d_v \cdot T_4}$, which will be introduced in the next subsection. By stacking dilated causal convolution layers, the model's receptive field grows exponentially, allowing for longer time series with fewer layers and thereby reducing computation costs. We validate the efficiency of dilated convolutional operations in our experiments (see Subsection 4.2) by comparing their computational costs with those of VAR-based methods.

To learn the causal relationship among system entities, we aggregate information from neighbors via a graph neural network (i.e., GraphSAGE (Hamilton et al., 2017)) and mimic fault propagation through a message-passing mechanism:

$$\tilde{\boldsymbol{X}}_v^0 = \sigma_2(\mathcal{A}_{old}(\hat{\boldsymbol{O}}_v^0 \oplus \boldsymbol{N}_v^0)\boldsymbol{W}^1), \text{where } \boldsymbol{N}_v^0[j] = \frac{1}{|\mathcal{N}_j|} \sum_{k \in N_j} \hat{\boldsymbol{O}}_v^0[k] \tag{6}$$

$$\tilde{\boldsymbol{X}}_v^i = \sigma_2((\mathcal{A}_{old} + \Delta\mathcal{A}_v)(\hat{\boldsymbol{O}}_v^i \oplus \boldsymbol{N}_v^i)\boldsymbol{W}^2), \text{where } \boldsymbol{N}_v^i[j] = \frac{1}{|\mathcal{N}_j|} \sum_{k \in N_j} \hat{\boldsymbol{O}}_v^i[k] \tag{7}$$

where $\boldsymbol{W}^1$ and $\boldsymbol{W}^2$ are weight matrices, $\oplus$ denotes concatenation, $\mathcal{N}_j$ represents node entity $j$'s neighbors, $\boldsymbol{N}_v^i$ aggregates neighbor information, $\mathcal{A}_{old}$ is the previous batch's learned causal graph, and $\Delta\mathcal{A}_v \in \mathbb{R}^{n \times n}$ is a learnable adjacency matrix. Unlike $\mathcal{A}_{old}$, $\Delta\mathcal{A}_v$ captures unique patterns in the current streaming data batch. $\tilde{\boldsymbol{X}}_v^0$ ($\tilde{\boldsymbol{X}}_v^i$) predicts future values based on previous lagged data $\hat{\boldsymbol{X}}_v^0$ ($\hat{\boldsymbol{X}}_v^i$), leveraging temporal dependencies captured by dilated convolutional neural networks. Finally, we minimize forecasting errors as follows:

$$\mathcal{L}_{temporal} = \frac{1}{n(d_L + d_M)} \sum_v \sum_{j=1}^{n} \sum_{k=1}^{d_v} [||\hat{\boldsymbol{X}}_v^0[j, k] - \tilde{\boldsymbol{X}}_v^0[j, k]||^2 + ||\hat{\boldsymbol{X}}_v^i[j, k] - \tilde{\boldsymbol{X}}_v^i[j, k]||^2] \tag{8}$$

Notice that leveraging Eq. 8 and the message passing mechanism of GNNs in Eq. 6 allows the model to encode causality in the learned adjacency matrix $\tilde{\boldsymbol{A}} = \mathcal{A}_{old} + \Delta\mathcal{A}_v$, such as $\boldsymbol{X} \rightarrow \boldsymbol{y}$, where $\boldsymbol{X}$ is a potential root cause and $\boldsymbol{y}$ is a Key Performance Indicator (KPI). Additionally, we add the trace exponential function $h(\tilde{\boldsymbol{A}}) = (tr(e^{\tilde{\boldsymbol{A}} \odot \tilde{\boldsymbol{A}}}) - n) = 0$ as a regularization term to ensure that $\tilde{\boldsymbol{A}}$ is acyclic (Pamfil et al., 2020), where $\odot$ denotes the Hadamard product of two matrices.

## 3.3 REPRESENTATION LEARNING WITH MULTI-FACTOR ATTENTION

In microservice systems, each system entity has multiple entity metrics and various log attributes/indicators, including CPU usage, memory usage, log frequency, log golden signal, etc.

Existing RCA methods analyze abnormal patterns from each factor (*i.e.*, metric or log indicator) individually, neglecting potential relationships among them. However, the importance of factors varies depending on the abnormal patterns. Hence, reassessing the contribution of each factor to causal structure learning is crucial. To bridge this gap, we propose to explore the correlation of different factors from two modalities and then assess the contribution of each factor to causal structure learning with the attention mechanism (Lu et al., 2016; Vaswani et al., 2017). Given two representations $\boldsymbol{H}_L^0$ and $\boldsymbol{H}_M^0$ in Eq. 3, we compute the multi-factor similarity matrix $\boldsymbol{C}_j^0 \in \mathbb{R}^{d_M \times d_L}$ for historical representation of the $j^{\text{th}}$ system entity to capture the correlation of different modalities and the relationship among multiple metrics and log indicators as follows:

$$\boldsymbol{C}_j^0 = \tanh\left(\boldsymbol{H}_M^0[j]\boldsymbol{W}^3(\boldsymbol{H}_L^0[j])^T\right) \tag{9}$$

where $\boldsymbol{W}^3 \in \mathbb{R}^{T_3 \times T_3}$ is a weight matrix and $\boldsymbol{H}_v^0[j]$ denotes the historical representation of the $j^{\text{th}}$ system entity for modality $v \in \{M, L\}$. Starting here, we will skip equations related to the $i^{\text{th}}$ batch of streaming data for brevity, unless their computation differs from historical data. This matrix measures the similarities between modalities and among multiple factors. By leveraging this similarity matrix, we aim to encode information from both modalities in the hidden representation $\boldsymbol{H}_v^0$ and assess the importance of each factor across both modalities, formulated as:

$$\boldsymbol{Z}_L^0[j] = \tanh\left(\boldsymbol{H}_L^0[j]\boldsymbol{W}^4 + \boldsymbol{H}_M^0[j]\boldsymbol{C}_j^0\boldsymbol{W}^5\right)$$
$$\boldsymbol{Z}_M^0[j] = \tanh\left(\boldsymbol{H}_M^0[j]\boldsymbol{W}^5 + \boldsymbol{H}_L^0[j](\boldsymbol{C}_j^0)^T\boldsymbol{W}^4\right)$$
$$\boldsymbol{a}_L^0[j] = \text{softmax}(\boldsymbol{w}^6\boldsymbol{Z}_L^0[j]), \quad \boldsymbol{a}_M^0[j] = \text{softmax}(\boldsymbol{w}^7\boldsymbol{Z}_M^0[j]) \tag{10}$$

where $\boldsymbol{W}^4 \in \mathbb{R}^{T_3 \times T_3}$ and $\boldsymbol{W}^5 \in \mathbb{R}^{T_3 \times T_3}$ are two weight matrices and $\boldsymbol{w}^6 \in \mathbb{R}^{T_4}$ and $\boldsymbol{w}^7 \in \mathbb{R}^{T_4}$ are two weight vectors. Notice that $\boldsymbol{a}_L^0[j]$ and $\boldsymbol{a}_M^0[j]$ measure the importance of each factor by encoding information from both modalities, capturing rich relationships for multi-modal and multi-dimensional data. Using these attention vectors, we encode all information learned from multiple factors of two modalities into the weighted representation $\hat{\boldsymbol{H}}_v^0 \in \mathbb{R}^{n \times T_3}$ by:

$$\hat{\boldsymbol{H}}_v^0[j] = \sum_{k=1}^{d_v} \boldsymbol{a}_v^0[i,k] \cdot \boldsymbol{H}_v^0[j,k] \tag{11}$$

After encoding the relationship among different factors and two modalities into $\hat{\boldsymbol{H}}_v^0$ and $\hat{\boldsymbol{H}}_v^i$, we aim to recover the factors of two modalities by:

$$\boldsymbol{O}_v^0 = \text{MLP}^0(\hat{\boldsymbol{H}}_v^0), \boldsymbol{O}_v^i = \text{MLP}^1(\hat{\boldsymbol{H}}_v^i) \tag{12}$$

where $\text{MLP}^0$ and $\text{MLP}^1$ are two multi-layer perceptrons (MLP) to recover the metrics, $\boldsymbol{O}_v^0 \in \mathbb{R}^{n \times d_v \times T_3}$ and $\boldsymbol{O}_v^i \in \mathbb{R}^{n \times d_v \times T_4}$. Here, we reshape $\boldsymbol{O}_v^0$ and $\boldsymbol{O}_v^i$, so that $\boldsymbol{O}_v^0 \in \mathbb{R}^{n \times d_v \cdot T_3}$ and $\boldsymbol{O}_v^i \in \mathbb{R}^{n \times d_v \cdot T_4}$. Overall, Eqs. 9, 10, 11 and 12 are combined to derive $MFL(\cdot)$ in Eq. 5. After we assess the contribution of each factor to the causal structure learning, we use $\boldsymbol{a}_v[j,k]$ to reweigh the importance of different factors in the future value prediction task in Eq. 8 and further encourage that the representations $\hat{\boldsymbol{H}}_v^0$ and $\hat{\boldsymbol{H}}_v^i$ should contain more information for the factor with a larger weight $\boldsymbol{a}_v[j,k]$. Therefore, Eq. 8 can be updated as follows:

$$\mathcal{L}_{temporal} = \frac{1}{n(d_L + d_M)} \sum_v \sum_{j=1}^n \sum_{k=1}^{d_v} [\boldsymbol{a}_v^0[j,k]||\hat{\boldsymbol{X}}_v^0[j,k] - \tilde{\boldsymbol{X}}_v^0[j,k]||^2$$
$$+ \boldsymbol{a}_v^i[j,k]||\hat{\boldsymbol{X}}_v^i[j,k] - \tilde{\boldsymbol{X}}_v^i[j,k]||^2] \tag{13}$$

### 3.4 GRAPH FUSION WITH CONTRASTIVE MULTI-MODAL LEARNING

To tackle the challenges of multi-modal learning (as discussed in challenge **C3** in Section 1), we propose to maximize the relatedness between two modalities via **contrastive mutual information maximization**. Given the representations of historical data $\hat{\boldsymbol{H}}_v^0$ and streaming data $\hat{\boldsymbol{H}}_v^i$ extracted from both metric and log data, we maximize the mutual information between these two modalities:

$$\mathcal{L}_{MI} = \mathcal{I}_\phi(\hat{\boldsymbol{H}}_M^0, \hat{\boldsymbol{H}}_L^0) + \mathcal{I}_\phi(\hat{\boldsymbol{H}}_M^i, \hat{\boldsymbol{H}}_L^i) \tag{14}$$

where $\mathcal{I}_\phi$ is the mutual information parameterized by a neural network $\phi$. Following InfoNCE style contrastive loss (Oord et al., 2018), we approximate the mutual information with its lower bound as follows:

$$\mathcal{I}_\phi(\hat{\boldsymbol{H}}_M^0, \hat{\boldsymbol{H}}_L^0) := \frac{1}{n} \sum_{j=1}^n \log \frac{\text{sim}(\phi(\hat{\boldsymbol{H}}_M^0[j]), \phi(\hat{\boldsymbol{H}}_L^0[j]))}{\sum_k \text{sim}(\phi(\hat{\boldsymbol{H}}_M^0[j]), \phi(\hat{\boldsymbol{H}}_L^0[k]))} \tag{15}$$

where $\text{sim}(a, b) = \exp(\frac{ab^T}{|a||b|})$ is the exponential of cosine similarity measurement between two entity representations $a$ and $b$.

To generate the causal graph for the current batch of data, simple addition may not work because it may result in dense and cyclical graphs. This issue might even exacerbate in the low-quality modality scenario, as one modality might convey more important information than others. The low-quality modality usually obscures the crucial patterns for causal graph learning if both modalities are treated with equal importance. To address this issue, we propose to measure the importance of two modalities with the correlation of multiple metrics captured in Eq. 9. Based on the similarity map for the current batch (*i.e.*, $\boldsymbol{C}_j^i$), we further measure the importance of each modality and fuse two causal graphs:

$$s_M = \frac{\sum_{j=1}^n \sum_{l=1}^{d_M} \exp(\sum_{k=1}^{d_L} \boldsymbol{C}_j^i[l, k])}{\sum_{j=1}^n \sum_{l=1}^{d_M} \exp(\sum_{k=1}^{d_L} \boldsymbol{C}_j^i[l, k]) + \sum_{j=1}^n \sum_{l=1}^{d_L} \exp(\sum_{l=1}^{d_M} \boldsymbol{C}_j^i[l, k])} \tag{16}$$

$$\mathcal{A} = (1 - s_M) \cdot (\mathcal{A}_{old} + \Delta\mathcal{A}_L) + s_M \cdot (\mathcal{A}_{old} + \Delta\mathcal{A}_M) \tag{17}$$

**Optimization.** The final objective function is written as:

$$\mathcal{L}_{sparse} = ||\Delta\mathcal{A}_L||_1 + ||\Delta\mathcal{A}_M||_1$$
$$\mathcal{L} = -\mathcal{L}_{MI} + \lambda_1 \mathcal{L}_{temporal} + \lambda_2 \mathcal{L}_{sparse} + \lambda_3 h(\mathcal{A}) \tag{18}$$

where $|| \cdot ||_1$ is the sparsity constraint imposed on the adjacency matrix and $\mathcal{L}_{sparse}$ aims to ensure that the changes of the edges are expected to be sparse. The trace exponential function $h(\mathcal{A}) = (tr(e^{\mathcal{A} \odot \mathcal{A}}) - n) = 0$ holds if and only if $\mathcal{A}$ is acyclic (Pamfil et al., 2020), where $\odot$ denotes the Hadamard product of two matrices. $\lambda_1$, $\lambda_2$ and $\lambda_3$ are the positive constant hyper-parameters.

## 3.5 NETWORK PROPAGATION-BASED ROOT CAUSE IDENTIFICATION

The propagation of malfunction effects from the root cause to adjacent entities implies that the immediate neighbors of system KPIs may not necessarily be the root causes themselves. To identify the root cause, we initially derive the transition probability matrix based on the causal graph $\mathcal{G}$ and then utilize a random walk with restart method (Tong et al., 2006) to simulate the spread patterns of malfunctions as follows:

$$\boldsymbol{P}_{ij} = \frac{\beta \mathcal{A}_{j,i}}{\sum_{k=1}^n \mathcal{A}_{k,i}} \tag{19}$$

The transition probability matrix $\boldsymbol{P}$ is the normalized adjacency matrix signified by the coefficient $\beta \in [0, 1]$. During the visiting exploration process, we may restart from the KPI node to revisit other system entities with the probability $c \in [0, 1]$. The equation for the random walk with restart is formulated by:

$$\boldsymbol{r}_{t+1} = (1 - c)\boldsymbol{P}\boldsymbol{r}_t + c\boldsymbol{r}_0 \tag{20}$$

where $\boldsymbol{r}_t$ represents the jumping probability at the $t^{\text{th}}$ step, $\boldsymbol{r}_0$ denotes the initial starting probability, and $c \in [0, 1]$ stands for the restart probability. Upon convergence of the jumping probability $\boldsymbol{r}_t$, the probability scores of the nodes are employed to rank the system entities and the top $k$ entities are selected as the most probable root causes for system failure.

**Stopping Criterion.** As the number of new data batches increases, the identified causal structure and its associated root cause list may gradually converge. To prevent unnecessary consumption of computing resources, we employ them as indicators for automatic termination of the online RCA process. We use the rank-biased overlap metric (RBO) (Webber et al., 2010) to measure the similarity between two root cause lists, effectively capturing the evolving trend of root cause rankings. Given the rank lists from the previous and current batches, denoted as $R_{t-1}$ and $R_t$ respectively, we quantify the similarity between these lists as follows:

$$\gamma = \text{RBO}(R_{t-1}, R_t) \tag{21}$$

where $\gamma \in [0, 1]$. A higher value of $\gamma$ indicates a greater similarity between the two root cause lists. The online RCA process is terminated when the similarity score $s$ surpasses a predefined threshold.

## 4 EXPERIMENTS

In this section, we evaluate the effectiveness of our proposed OCEAN by comparing it with state-of-the-art root cause analysis techniques. Additionally, we conduct a case study and an ablation study to further validate the assumptions outlined in the previous sections.

### 4.1 EXPERIMENTAL SETUP

**Datasets**. We evaluate the performance of OCEAN using three public real-world datasets: (1) **Product Review*** (Zheng et al., 2024b): A microservice system dedicated to online product reviews, encompassing 234 pods deployed across 6 cloud servers. It recorded four system faults between May 2021 and December 2021. (2) **Online Boutique**† (Yu et al., 2023): A microservice system designed for e-commerce, including five system faults. (3) **Train Ticket** (Yu et al., 2023): A microservice system for railway ticketing services, also with five system faults. All three datasets contain two modalities: system metrics and system logs.

**Evaluation Metrics**. We choose four widely-used metrics (Wang et al., 2023c; Meng et al., 2020): (1) **Precision@K (PR@K)**: Measures the accuracy of the top-K predicted root causes. (2) **Mean Average Precision@K (MAP@K)**: Evaluates the overall accuracy of the top-K predicted causes. (3) **Mean Reciprocal Rank (MRR)**: Assesses the ranking capability of the models. (4) **Time**: Measures the training time (in seconds) for each batch of data. Details on the first three metrics are provided in Appendix D and we provide the time complexity analysis in Appendix B.

**Baselines**. We compare OCEAN with seven causal discovery based RCA methods: (1) **PC** (Burr, 2003): A classic constraint-based algorithm that identifies the causal graph's skeleton using an independence test. (2) **Dynotears** (Pamfil et al., 2020): Constructs dynamic Bayesian networks through vector autoregression models. (3) **C-LSTM** (Tank et al., 2022): Utilizes LSTM to model temporal dependencies and capture nonlinear Granger causality. (4) **GOLEM** (Ng et al., 2020): Relaxes the hard Directed Acyclic Graph (DAG) constraint in NOTEARS (Zheng et al., 2018) with a scoring function. (5) **REASON** (Wang et al., 2023c): Learns both intra-level and inter-level causal relationships in interdependent networks. (6) **MULAN** (Zheng et al., 2024a): A multi-modal method that captures both modality-invariant and modality-specific representations. (7) **CORAL** (Wang et al., 2023a): A VAR-based online RCA method that decouples state-invariant and state-dependent information.

The first four baseline models were originally designed to learn causal structures solely from time series data. As outlined in (Wang et al., 2023c;a), these causal discovery models can be extended to identify the root cause nodes. In this process, we first apply causal discovery models to learn the causal graphs, then utilize random walk with restarts (Wang et al., 2023a) on these graphs to identify the top $K$ nodes as root causes.

### 4.2 PERFORMANCE EVALUATION

**Experimental Results.** In this subsection, we present the performance evaluation results in Tables 2, 3, and 5 for various methods. Due to the page limit, we have moved the results for the Train Ticket dataset (*i.e.*, Table 5) to Appendix E.1. Notably, although many baseline methods (*e.g.*, PC, C-LSTM, REASON, Dynotears, GOLEM) are tailored for single-modal scenarios, we assess their performance in both single-modal contexts (*e.g.*, system metrics only or system logs only) and multi-modal scenarios. System logs are considered additional system metrics, enabling these single-modal methods to be evaluated in a multi-modal context. We derive an average ranking score from different system metrics as the final result for all single-modal methods.

Our findings include: (1) Most baseline methods show improved performance when leveraging multi-modal data across three datasets. (2) CORAL, as an online RCA method, surpasses all offline methods across seven metrics. (3) OCEAN consistently outperforms all baselines across the datasets. (4) Both CORAL and OCEAN demonstrate shorter training time compared to offline methods, with OCEAN reducing its computational costs to 1/9 that of CORAL on the Product Review dataset. This reduction is credited to the efficiency of dilated convolutional operations and the design of the

---

*https://lemma-rca.github.io/docs/data.html
†https://github.com/IntelligentDDS/Nezha

Table 2: Results on Product Review dataset w.r.t different metrics.

| Modality | Model | PR@1 | PR@5 | PR@10 | MRR | MAP@3 | MAP@5 | MAP@10 | Time (s) |
|---|---|---|---|---|---|---|---|---|---|
| Metric Only | PC | 0 | 0 | 0 | 0.034 | 0 | 0 | 0 | 225.19 |
| | Dynotears | 0 | 0.25 | 0.50 | 0.092 | 0 | 0.05 | 0.175 | 390.37 |
| | C-LSTM | 0.25 | 0.5 | 0.5 | 0.409 | 0.417 | 0.45 | 0.475 | 1482.01 |
| | GOLEM | 0 | 0 | 0.25 | 0.043 | 0 | 0 | 0.025 | 308.25 |
| | REASON | 0.25 | **1.0** | **1.0** | 0.5625 | 0.583 | 0.75 | 0.875 | 247.87 |
| | CORAL | 0.5 | **1.0** | **1.0** | 0.75 | 0.833 | 0.9 | 0.95 | 146.46 |
| Log Only | PC | 0 | 0 | 0 | 0.043 | 0 | 0 | 0 | 93.98 |
| | Dynotears | 0 | 0 | 0.25 | 0.058 | 0 | 0 | 0.075 | 142.26 |
| | C-LSTM | 0 | 0 | 0.25 | 0.059 | 0 | 0 | 0.075 | 602.92 |
| | GOLEM | 0 | 0 | 0.25 | 0.058 | 0 | 0 | 0.075 | 144.8 |
| | REASON | 0 | 0 | 0.5 | 0.088 | 0 | 0 | 0.1 | 129.17 |
| | CORAL | 0 | 0 | 0.5 | 0.118 | 0 | 0 | 0.2 | 50.29 |
| Multi-Modality | PC | 0 | 0 | 0.25 | 0.054 | 0 | 0 | 0.075 | 300.26 |
| | Dynotears | 0 | 0.25 | 0.5 | 0.114 | 0 | 0.05 | 0.225 | 426.78 |
| | C-LSTM | 0.25 | 0.5 | 0.5 | 0.341 | 0.25 | 0.35 | 0.425 | 1808.76 |
| | GOLEM | 0 | 0 | 0.25 | 0.066 | 0 | 0 | 0.05 | 452.25 |
| | REASON | 0.5 | **1.0** | **1.0** | 0.687 | 0.667 | 0.8 | 0.9 | 303.5 |
| | MULAN | 0.75 | **1.0** | **1.0** | 0.833 | 0.833 | 0.9 | 0.95 | 255.74 |
| | CORAL | 0.75 | **1.0** | **1.0** | 0.875 | 0.917 | 0.95 | 0.975 | 186.73 |
| | OCEAN | **1.0** | **1.0** | **1.0** | **1.0** | **1.0** | **1.0** | **1.0** | **20.16** |

Table 3: Results on Online Boutique w.r.t different metrics.

| Modality | Model | PR@1 | PR@3 | PR@5 | MRR | MAP@2 | MAP@3 | MAP@5 | Time (s) |
|---|---|---|---|---|---|---|---|---|---|
| Metric Only | PC | 0.2 | 0.4 | 0.6 | 0.39 | 0.3 | 0.333 | 0.4 | 5.25 |
| | Dynotears | 0.2 | 0.4 | 0.4 | 0.344 | 0.2 | 0.267 | 0.32 | 14.56 |
| | C-LSTM | 0 | 0.4 | 0.8 | 0.3 | 0.1 | 0.2 | 0.44 | 20.75 |
| | GOLEM | 0 | 0.4 | 0.6 | 0.291 | 0.2 | 0.267 | 0.36 | 4.32 |
| | REASON | 0.4 | 0.8 | **1.0** | 0.617 | 0.5 | 0.6 | 0.76 | 3.23 |
| | CORAL | 0.2 | **1.0** | **1.0** | 0.6 | 0.6 | 0.733 | 0.84 | 2.99 |
| Log Only | PC | 0 | 0.4 | 0.6 | 0.257 | 0.1 | 0.2 | 0.32 | 3.88 |
| | Dynotears | 0 | 0.2 | 0.6 | 0.207 | 0 | 0.067 | 0.24 | 10.23 |
| | C-LSTM | 0 | 0.4 | 0.6 | 0.267 | 0.1 | 0.2 | 0.36 | 15.07 |
| | GOLEM | 0 | 0.4 | 0.8 | 0.248 | 0 | 0.133 | 0.36 | 3.39 |
| | REASON | 0.2 | 0.8 | 0.8 | 0.458 | 0.3 | 0.467 | 0.6 | 2.39 |
| | CORAL | 0.2 | 0.6 | **1.0** | 0.457 | 0.3 | 0.4 | 0.6 | 2.04 |
| Multi-Modality | PC | 0.4 | 0.8 | **1.0** | 0.573 | 0.4 | 0.533 | 0.68 | 6.78 |
| | Dynotears | 0.2 | 0.6 | **1.0** | 0.467 | 0.3 | 0.4 | 0.64 | 16.38 |
| | C-LSTM | 0.2 | 0.4 | **1.0** | 0.45 | 0.3 | 0.333 | 0.6 | 22.66 |
| | GOLEM | 0.2 | 0.6 | **1.0** | 0.467 | 0.3 | 0.4 | 0.64 | 5.68 |
| | REASON | 0.4 | **1.0** | **1.0** | 0.667 | 0.6 | 0.733 | 0.84 | 4.51 |
| | MULAN | 0.4 | 0.8 | **1.0** | 0.617 | 0.5 | 0.6 | 0.76 | 4.96 |
| | CORAL | 0.4 | **1.0** | **1.0** | 0.7 | 0.7 | 0.8 | 0.88 | 3.63 |
| | OCEAN | **0.6** | **1.0** | **1.0** | **0.8** | **0.8** | **0.867** | **0.92** | **1.84** |

multi-factor attention module. CORAL's approach of individually learning and then fusing causal graphs for each metric is computationally intensive in an online setting. Furthermore, OCEAN shows a notable improvement in MRR on the Product Review dataset, outperforming CORAL by 12.5%. Additionally, OCEAN exceeds CORAL by 20% in PR@1 and 10% in MAP@2 on the Online Boutique dataset, benefiting from the assessment of the importance of multiple factors and exploring correlations among different modalities.

**Ablation Study.** In this subsection, we evaluate the effectiveness of individual components within the objective function of OCEAN (Eq. 18). Specifically, we define OCEAN-F and OCEAN-M as variants that lack the multi-factor attention learning module and the contrastive multi-modal learning module, respectively, while OCEAN-S removes the sparse constraint. The results, shown in

Table 4: Ablation study on three datasets w.r.t MRR.

| Model | Product Review | Online Boutique | Train Ticket |
|---|---|---|---|
| OCEAN | **1.0** | **0.8** | **0.381** |
| OCEAN-F | 0.75 | **0.8** | 0.331 |
| OCEAN-M | 0.875 | 0.7 | 0.320 |
| OCEAN-S | 0.833 | 0.7 | 0.345 |

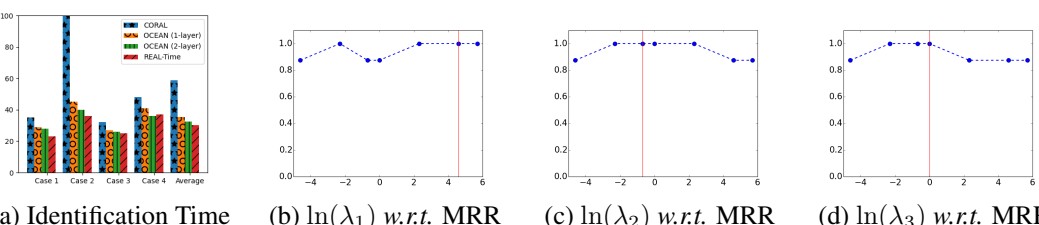

(a) Identification Time    (b) $\ln(\lambda_1)$ *w.r.t.* MRR    (c) $\ln(\lambda_2)$ *w.r.t.* MRR    (d) $\ln(\lambda_3)$ *w.r.t.* MRR

Figure 2: Figure (a) shows the identification time for four cases as well as the average identification time. Figures (b), (c), and (d) are the parameter analysis on the Product Review dataset w.r.t MRR.

Table 4, indicate a significant performance degradation when any component is omitted. Specifically, removing the multi-factor attention module results in 25% and 5% performance drop on the Product Review dataset and Train Ticket dataset, respectively. Eliminating the contrastive multi-modal learning module leads to 12.5% reduction on the Product Review dataset. These findings underscore the importance of each component in maintaining OCEAN's high performance. Additionally, as detailed in Appendix E.2, replacing the dilated convolutional network with LSTM or Transformer model resulted in performance decline, reinforcing the effectiveness of our design.

**Case Study.** In this subsection, we evaluate the promptness of two online RCA methods on the Product Review dataset, CORAL and OCEAN, as shown in Figure 2 (a). Note that we also evaluate the effectiveness of the long-term temporal causal structure learning module by varying the number of dilated convolutional layers in OCEAN, specifically comparing configurations with one and two layers. In Figure 2 (a), the y-axis represents the batch index at which an RCA method meets the stopping criteria, and the real-time marker indicates the actual system failure time. A lower batch index value signifies faster identification of the ground-truth root cause by the RCA method. Notably, CORAL did not successfully rank the ground-truth root cause first in case 2, so we use the total number of batches to represent its detection time for a fair comparison. Our observations reveal that CORAL experiences about a 10-epoch delay relative to real-time in most cases, whereas OCEAN (2-layer) achieves quicker detection than OCEAN (1-layer). This improvement confirms our hypothesis that adding more dilated convolutional layers enhances the model's ability to capture longer temporal dependencies, as discussed in Subsection 3.2.

**Parameter Analysis.** In this subsection, we present detailed parameter sensitivity analysis conducted on the Product Review dataset, with additional analyses for other datasets available in Appendix E.3. Specifically, we explore the impact of three parameters, $\lambda_1$, $\lambda_2$, and $\lambda_3$, on the overall objective functions as defined in Eq. 18. The experimental results are displayed in Figures 2 (b), (c), and (d), showing the Mean Reciprocal Rank (MRR) on the Product Review dataset. On these figures, the x-axis represents $\ln(\lambda_i)$ for $i \in [1, 2, 3]$, and the y-axis shows the MRR score. Our analysis reveals that a higher $\lambda_1$ (*e.g.*, $\ln(\lambda_1) = 4.6$) significantly enhances performance, underscoring the vital role of the long-term temporal causal structure learning module in capturing temporal dependencies among system entities. Conversely, $\lambda_2$ and $\lambda_3$ exhibit optimal performance at relatively lower values (*e.g.*, $\ln(\lambda_2) = -0.7$ and $\ln(\lambda_3) = 0$), with performance declining noticeably at higher levels. However, further reducing $\lambda_2$ and $\lambda_3$ also leads to diminished performance, which verifies the important role of sparse regularization and the acyclic constraint of the causal graph.

## 5 CONCLUSION

In this paper, we investigate the challenging problem of online multi-modal root cause localization in microservice systems. We introduce OCEAN, a novel online causal structure learning framework designed to effectively identify root causes using diverse data sources. OCEAN utilizes a dilated convolutional neural network to capture long-term temporal dependencies and employs graph neural networks to establish causal relationships among system entities and key performance indicators. Additionally, we develop a multi-factor attention mechanism to evaluate and refine the contributions of various factors to the causal graph. Furthermore, OCEAN incorporates a contrastive mutual information maximization-based graph fusion module to enhance interactions between different modalities and optimize their mutual information. The effectiveness of OCEAN is validated through extensive experiments on three real-world datasets, demonstrating its robustness and efficiency.

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

## A    MORE RATIONAL BEHIND THE MODEL DESIGN

The main issue of Recurrent Neural Networks (RNNs) and Transformers is their high computational cost, which makes them impractical for online root cause analysis. We chose Dilated Convolutional Neural Networks (DCNNs) for their efficiency and ability to capture long-range dependencies, both of which are essential in online settings where computational efficiency is critical. Unlike RNNs and Transformers, which could capture temporal dependencies but often come with high computational overhead, DCNNs are parallelizable and offer a lower time complexity of $O(NTkC)$, where $T$ is the sequence length, $L$ is the number of layers, $C$ is the number of filters, and $k$ is the filter size. In contrast, the complexity of the Transformer is $O(LT^2d)$, where $T$ is the sequence length, $L$ is the number of layers, and $d$ is the hidden feature dimensionality. As a result, Transformers are computationally more expensive than dilated convolutional neural networks. We further demonstrate the efficiency and effectiveness of our method compared to LSTM and Transformers in the ablation study presented in Table 6.

The rational behind the multi-factor attention mechanism is to capture complex relationships among various factors (*e.g.*, CPU usage and memory usage metrics in a microservice system) across two modalities. A common limitation of existing methods is their focus on correlations between modalities, often overlooking the importance of individual factors within each modality. Our mechanism addresses this by reassessing the significance of each factor, enhancing causal structure learning. This enables our model to dynamically adjust to the changing significance of factors, which is crucial for accurate root cause analysis.

We integrate contrastive learning into our model because of its proven effectiveness in multi-modal tasks. It extracts shared information across different modalities, enhancing robustness and addressing challenges posed by low-quality data. By assigning weights to each modality based on its importance, contrastive learning ensures that critical patterns are not obscured by noise. This approach helps maintain the quality of causal graph learning, even when dealing with data of varying quality.

## B    TIME COMPLEXITY

The time complexity of dilated convolution based causal structure learning is $O(NTkC)$. Here, $T$ is the sequence length, $L$ is the number of layers, $C$ is the number of filters, and $k$ is the filter size. The time complexity of multi-factor attention module is $O(d_M d_L d)$, where $d$ is the hidden feature dimensionality, and $d_M$ and $d_L$ are the number of factors in two modalities. The time complexity of contrastive learning module is $O(n^2 d)$, where $n$ is the number of system entities.

## C    LOG FEATURE/INDICATOR EXTRACTION

In this subsection, we provide the details of converting the raw data into the time series, though this is not within the scope of this work. Specifically, we first use the Drain parser He et al. (2017) to transform the unstructured log event into structured log templates for each entity. Then, we partition the log data with fixed time windows, such as 5 minutes, and set time steps at 10 seconds. Within these intervals, we count the occurrence of each log template to derive the log frequency feature. The extraction of the log frequency feature is inspired by the insight that the recurrence of a log event template often correlates with its significance. For instance, when a microservice system experiences Distributed Denial of Service (DDoS) attacks, the system will produce an unusual volume of system logs, indicating abnormal activity. Thus, the log frequency provides the information to identify unusual patterns indicative of potential failure scenarios. In addition to log frequency, we also extract a second type of log feature known as the 'golden signal.' Notice that different from log frequency, golden signal heavily relies on domain knowledge and it only focuses on the abnormal system logs. More specifically, we are only interested in some keywords, including 'error,' 'exception,' 'critical,' 'fatal', and various others indicative of system anomalies. By identifying these keywords within log event templates, we can discern abnormal occurrences for system failure localization. Similar to the frequency-based feature, we compute the number of abnormal log events to derive the golden signal-based feature.

# D    EVALUATION METRICS

We evaluate the model performance with the following four widely-used metrics Meng et al. (2020):

(1). **Precision@K (PR@K)**: It measures the probability that the top $K$ predicted root causes are relevant by:

$$PR@K = \frac{1}{|\mathbb{A}|} \sum_{a \in \mathbb{A}} \frac{\sum_{i<k} R_a(i) \in V_a}{\min(K, |v_a|)} \tag{22}$$

where $\mathbb{A}$ is the set of system faults, $a$ is one fault in $\mathbb{A}$, $V_a$ is the real root causes of $a$, $R_a$ is the predicted root causes of $a$, and i is the $i$-th predicted cause of $R_a$.

(2). **Mean Average Precision@K (MAP@K)**: It evaluates the top $K$ predicted causes from the overall perspective formulated as:

$$MAP@K = \frac{1}{K|\mathbb{A}|} \sum_{a \in \mathbb{A}} \sum_{i \leq j \leq K} PR@j \tag{23}$$

where a higher value indicates a better performance.

(3). **Mean Reciprocal Rank (MRR)**: It assesses the ranking capability of models, defined as:

$$PR@K = \frac{1}{|\mathbb{A}|} \sum_{a \in \mathbb{A}} \frac{1}{rank_{R_a}} \tag{24}$$

where $rank_{R_a}$ is the rank number of the first correctly predicted root cause for system fault $a$.

(4). **Time**: Measures the training time (in seconds) for each batch of data.

# E    ADDITIONAL EXPERIMENT

## E.1    EXPERIMENTAL RESULTS

We report the experimental results on Train Ticket dataset in Table 5.

Table 5: Results on Train Ticket w.r.t different metrics.

| Modality | Model | PR@1 | PR@5 | PR@10 | MRR | MAP@3 | MAP@5 | MAP@10 | Time (s) |
|---|---|---|---|---|---|---|---|---|---|
| Metric Only | PC | 0 | 0 | 0.2 | 0.067 | 0 | 0 | 0.06 | 9.65 |
| | Dynotears | 0 | 0 | 0 | 0.047 | 0 | 0 | 0 | 21.3 |
| | C-LSTM | 0 | 0.2 | 0.2 | 0.097 | 0 | 0.08 | 0.14 | 30.63 |
| | GOLEM | 0 | 0.2 | 0.2 | 0.098 | 0 | 0.08 | 0.14 | 9.56 |
| | REASON | **0.2** | **0.4** | 0.6 | 0.323 | 0.2 | **0.36** | 0.48 | 9.03 |
| | CORAL | 0 | **0.4** | **1.0** | 0.184 | 0 | 0.16 | 0.5 | 5.72 |
| Log Only | PC | 0 | 0.2 | 0.4 | 0.166 | 0.133 | 0.16 | 0.26 | 6.34 |
| | Dynotears | 0 | 0 | 0.2 | 0.072 | 0 | 0 | 0.02 | 17.26 |
| | C-LSTM | 0 | 0 | 0.2 | 0.072 | 0 | 0 | 0.02 | 21.40 |
| | GOLEM | 0 | 0.2 | 0.6 | 0.125 | 0 | 0.08 | 0.24 | 7.54 |
| | REASON | 0 | 0.2 | 0.6 | 0.126 | 0 | 0.08 | 0.28 | 7.58 |
| | CORAL | 0 | 0.2 | 0.8 | 0.138 | 0 | 0.08 | 0.32 | 3.35 |
| Multi-Modality | PC | 0 | 0 | 0.2 | 0.083 | 0 | 0 | 0.1 | 12.83 |
| | Dynotears | 0 | 0.4 | 0.6 | 0.141 | 0 | 0.16 | 0.32 | 27.82 |
| | C-LSTM | **0.2** | 0.4 | 0.6 | 0.294 | 0.2 | 0.28 | 0.36 | 36.76 |
| | GOLEM | 0 | 0.4 | 0.6 | 0.144 | 0 | 0.16 | 0.3 | 12.16 |
| | REASON | **0.2** | 0.4 | 0.6 | 0.300 | 0.2 | 0.28 | 0.42 | 12.81 |
| | MULAN | **0.2** | **0.4** | **1.0** | 0.317 | 0.2 | 0.28 | 0.46 | 11.42 |
| | CORAL | **0.2** | **0.4** | **1.0** | 0.334 | 0.2 | 0.28 | 0.56 | 7.26 |
| | OCEAN | **0.2** | **0.4** | **1.0** | **0.381** | **0.333** | **0.36** | **0.58** | **3.22** |

## E.2    ADDITIONAL ABLATION STUDY

We conduct additional ablation study by replacing the dilated convolutional neural network with LSTM or Transformer model on the Product Review and Train Ticket datasets in Table 6. The results showed a performance decline, demonstrating the effectiveness of our design.

Table 6: Additional Ablation Study

| Product Review Dataset | PR@1 | PR@5 | PR@10 | MRR | MAP@3 | MAP@5 | MAP@10 | Time (s) |
|---|---|---|---|---|---|---|---|---|
| OCEAN | 1 | 1 | 1 | 1 | 1 | 1 | 1 | 20.16 |
| OCEAN-LSTM | 0.25 | 1 | 1 | 0.542 | 0.583 | 0.75 | 0.875 | 546.37 |
| OCEAN-Transformer | 0.5 | 1 | 1 | 0.75 | 0.833 | 0.9 | 0.95 | 658.72 |
| **Train Ticket Dataset** | PR@1 | PR@5 | PR@10 | MRR | MAP@3 | MAP@5 | MAP@10 | Time (s) |
| OCEAN | 0.2 | 0.4 | 1.0 | 0.381 | 0.333 | 0.36 | 0.58 | 3.22 |
| OCEAN-LSTM | 0.2 | 0.6 | 0.8 | 0.342 | 0.2 | 0.36 | 0.54 | 12.6 |
| OCEAN-Transformer | 0.2 | 0.2 | 1 | 0.314 | 0.2 | 0.2 | 0.5 | 18.9 |

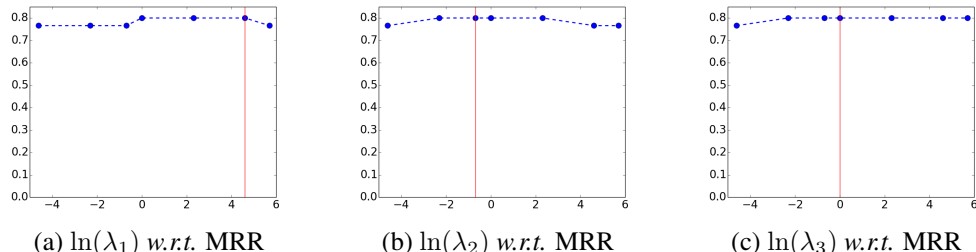

(a) $\ln(\lambda_1)$ *w.r.t.* MRR    (b) $\ln(\lambda_2)$ *w.r.t.* MRR    (c) $\ln(\lambda_3)$ *w.r.t.* MRR

Figure 3: Parameter analysis on the Online Boutique dataset w.r.t MRR.

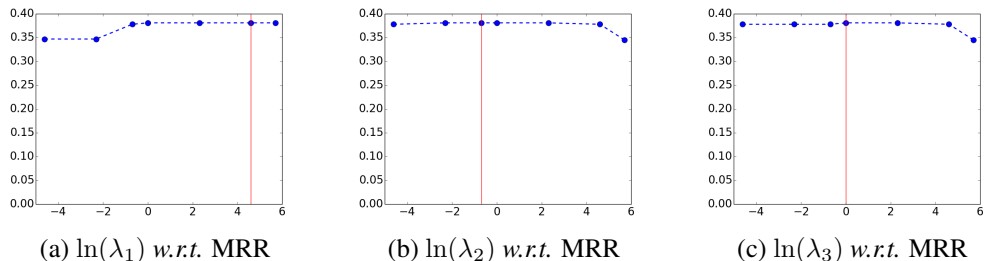

(a) $\ln(\lambda_1)$ *w.r.t.* MRR    (b) $\ln(\lambda_2)$ *w.r.t.* MRR    (c) $\ln(\lambda_3)$ *w.r.t.* MRR

Figure 4: Parameter analysis on the Train Ticket dataset w.r.t MRR.

Table 7: Experimental results with more modality

| Product Review Dataset | PR@1 | PR@5 | PR@10 | MRR | MAP@3 | MAP@5 | MAP@10 | Time (s) |
|---|---|---|---|---|---|---|---|---|
| OCEAN | 1 | 1 | 1 | 1 | 1 | 1 | 1 | 20.1 |
| OCEAN + trace | 1 | 1 | 1 | 1 | 1 | 1 | 1 | 26.3 |
| **Train Ticket** | PR@1 | PR@5 | PR@10 | MRR | MAP@3 | MAP@5 | MAP@10 | Time (s) |
| OCEAN | 0.2 | 0.4 | 1 | 0.38 | 0.33 | 0.36 | 0.58 | 3.2 |
| OCEAN + trace | 0.2 | 0.6 | 1 | 0.39 | 0.33 | 0.44 | 0.62 | 3.8 |

Table 8: Comparison with Physical Graph

| Graphs | SHD | AUROC |
|---|---|---|
| Causal graph Learned From Metric Data | 0.314 | 0.865 |
| Causal graph Learned From Log Data | 0.593 | 0.663 |
| Fused Causal Graph | 0.298 | 0.881 |

### E.3 ADDITIONAL PARAMETER ANALYSIS

In this subsection, we conduct a comprehensive parameter sensitivity analysis on the Online Boutique and Train Ticket datasets. Similarly, we assess the impact of three parameters on the overall objective functions (Eq. 18): $\lambda_1$, $\lambda_2$, and $\lambda_3$. Figures 3 and Figures 4 show the experimental results in terms of Mean Reciprocal Rank (MRR) on the Online Boutique and Train Ticket datasets. The x-axis represents $\ln(\lambda_i)$, where $i \in [1, 2, 3]$, and the y-axis indicates the MRR score. We consistently observe that OCEAN favors a larger value of $\lambda_1$ on these two datasets as the temporal causal structure learning module is crucial in capturing both temporal and causal dependency. Different from the parameter analysis on the Product Review dataset, we find that $\lambda_2$ and $\lambda_3$ are not very sensitive on the Online Boutique and Train Ticket datasets. We conjecture that this can be attributed to the small size of these two datasets and both sparse regularization and acyclic constraint contribute less to securing high performance on these two datasets than the Product Review dataset.

### E.4 EXPERIMENTAL RESULTS WITH MORE MODALITY

OCEAN can naturally extend to include additional modalities, such as traces. These types of data can enhance the model's ability to capture complex interactions and dependencies within the system. We conducted additional experiments by incorporating traces into the AIOps and Train Ticket datasets. The results demonstrated improved performance, as the inclusion of traces provided valuable context and enriched the causal structure learning. This additional information allows OCEAN to more accurately identify root causes and improve the precision of its analysis.

### E.5 COMPARISON WITH PHYSICAL GRAPH

Here, we evaluate the quality of the learned causal graph by comparing it with the physical dependency graph with two settings. In the first setting, we compared the causal graph learned by each modality (corresponding to the inter-modal graphs) and in the second setting, we compared the fused causal graph from two modality (corresponding to the intra-model graph). Following Dynotear Pamfil et al. (2020), we use AUROC and SHD as two metrics to quantify the difference between learned causal graphs and the physical dependency graph.

## F REPRODUCIBILITY

All experiments are conducted on a desktop running Ubuntu 18.04.5 with an Intel(R) Xeon(R) Silver 4110 CPU @2.10GHz and one 11GB GTX2080 GPU. In the experiment, we set the size of historical metric and log data to 8-hour intervals and each batch is set to be a 10-minute interval. We use the Adam as the optimizer and we train the model for 100 iterations at each batch. We use two layers of dilated convolutional operations in the experiment. As for the stopping criteria, we terminate the identification process if the similarity $\gamma$ between the current batch and the previous batch is greater than 0.9 for three consecutive times.

