# OpenReview forum: "OCEAN: Online Multi-modal Root Cause Analysis for Microservice Systems"
_ICLR.cc/2025/Conference — ICLR 2025 Conference Withdrawn Submission_

### Official Review · Reviewer_jmm7 · 2024-10-30

**Soundness:** 2
**Presentation:** 1
**Contribution:** 3
**Rating:** 3
**Confidence:** 2

**Summary:**

This paper presents OCEAN, an online multi-modal method for root cause analysis (RCA) in microservice systems. The approach utilizes various techniques, including a dilated convolutional neural network to capture long-term temporal relationships, multi-factor attention for encoding feature correlation, a graph neural network (GNN) for identifying causal relationships, and a random walk with revisiting on the derived causal graph for ranking root causes, etc.

**Strengths:**

- The problem of online multi-modal RCA is highly relevant and valuable.
- The performance of OCEAN appears to be exceptional.

**Weaknesses:**

- The notations used throughout the paper lack clarity.
- Several implementation details are omitted, such as the architecture of the MLP used in L_MI and the hyperparameters.

**Questions:**

1. The notations in the problem statement are not sufficiently clear, e.g.,  $n-1$ and $d_M$ in line 158. It would be helpful for the authors to specify the dimensions of the matrices involved. Additionally, do all system entities share the same "entity metrics" as features?

I would appreciate clarification on the following points:

- Is $n-1$ referring to the number of system "entities" or "entity metrics"?
- Is only one system KPI considered?  Is the bold symbol $\boldsymbol{y}$ simply a one-dimensional vector? I had assumed multiple KPIs could be monitored.
- I find it challenging to understand the replication of the KPI $d_M$ times to create a tensor version of $\hat{X}$. If all entities share the same "metrics," that makes sense; could you clarify this?
- The 1-D dilated convolution described in Eq. (2-5) is unclear. Is $\boldsymbol{f}(t)$ a scalar or a vector? Additionally, could you elaborate on Eq. (2-3) in relation to the tensor input x? What is the rationale for using "two" 1D kernels and activation functions (tanh, sigmoid) in Eq. (3)?

2. It seems that Eq. (8) or Eq. (13) represents the loss for only the i-th batch, yet the authors incorporate them into their final objective (18). What is the precise training procedure for the online framework?

3. If the goal is to maximize (14), then L_MI in (18) should have a negative sign.

4. In Figure 2 (b,c,d), the setting of other hyper-parameters should be revealed.
5. The proposed method is based on GCN for causal discovery, have you tried other GNN-based causal discovery methods?

---

> ### Author Response · Authors · 2024-11-22
> **Reply by Authors**
>
> Thank you for your invaluable feedback. We would like to address your primary concerns and provide a response below.
>
> - **The notations in the problem statement are not sufficiently clear, e.g., n-1 and d_M in line 158. Additionally, do all system entities share the same "entity metrics" as features? Is n-1 referring to the number of system "entities" or "entity metrics"?**
>
> A: Yes, all system entities share the same "entity metrics" as features and $n-1$ refers to the number of system entities. The reason why we use n-1 to denote the number of system entities is that we also include one KPI in our causal structure learning such that the total number of nodes in the adjacency matrix is n rather than n+1.
>
> - **Is only one system KPI considered?Is the bold symbol $\bm{y}$ simply a one-dimensional vector? I had assumed multiple KPIs could be monitored.**
>
> A: Yes, for simplicity, our method only considers one KPI. $\bm{y}$ is simply a one-dimensional vector. When multiple KPIs are available, we select one KPI that is mostly related to system fault based on domain knowledge.
>
> - **I find it challenging to understand the replication of the KPI $d_M$ times to create a tensor version of $\hat{\bm{X}}$. If all entities share the same "metrics," that makes sense; could you clarify this?**
>
> A: In the root cause analysis, KPI serves as the “label information” to guide the RCA methods to identify the potential root causes. The reason why we replicate the KPI $d_M$ times is to match the number of system metrics features and then use the same KPI to measure causal relation among different system entities and KPI for different system metrics features or different log features. Notice that When multiple KPIs are available, we select one KPI that is mostly related to system fault based on domain knowledge.
>
>  - **The 1-D dilated convolution described in Eq. (2-5) is unclear. Is f(t) a scalar or a vector? Could you elaborate on Eq. (2-3) in relation to the tensor input x? What is the rationale for using "two" 1D kernels and activation functions (tanh, sigmoid) in Eq. (3)?**
>
> A: f(t) is a scalar. It should be $f\in \mathbb{R}^{K}$ rather than f(t). We have corrected the typo in the updated version. The rationale for using two activation functions (tanh and sigmoid) is similar to the rationale of LSTM. The **sigmoid function** is used to control the extent to which the cell state should influence the output and the **tanh function** is applied to the cell state to constrain its range between -1 and 1, ensuring that the outputs remain stable and manageable. The **sigmoid** part decides "how much" information should be passed, while the tanh part provides a scaled and normalized version of the "what" information to pass.
>
> - **It seems that Eq. (8) or Eq. (13) represents the loss for only the i-th batch, yet the authors incorporate them into their final objective (18). What is the precise training procedure for the online framework?**
>
> A: In the online setting, every several minutes, a new batch data (e.g., i-th batch) is given. Once the online failure detection is triggered, starting from the i-th batch, we keep finetuning the model with equation 18. Our proposed method is not only fine tuned with the i-th batch but also the next several batches, which aims to ensure the correctness of the final results. We provide the stopping criteria in Section 3.5.
>
> - **If the goal is to maximize (14), then L_MI in (18) should have a negative sign.**
>
> A: We would like to point out that we add the negative sign in equation 15 rather than equation 18. We will take your suggestion and update both equation 15 and equation 18 to avoid the confusion.
>
> - **In Figure 2 (b,c,d), the setting of other hyper-parameters should be revealed.**
>
> A: The hyperparameters $\lambda_1$, $\lambda_2$ and $\lambda_3$ were selected from the set [0.01, 0.1, 0.5, 1, 10, 100, 300]. In Figure 2, the vertical red line in each subfigure indicates the specific parameter settings ($\lambda_1$, $\lambda_2$ and $\lambda_3$) used to achieve optimal performance on the AIOps dataset. Similarly, Figures 3 and 4 provide the parameter settings that yield the best results for the other two datasets. To enhance the reproducibility of our experimental results, we will further refine our parameter analysis and highlight the optimal parameter settings throughout the paper.

---

> > ### Author Response · Authors · 2024-11-22
> > **Reply by Authors**
> >
> > - **The proposed method is based on GCN for causal discovery, have you tried other GNN-based causal discovery methods?**
> >
> > A: Below are the additional experimental results with different types of GNN.
> > | Product Review	| PR@1 |  PR@5  |  PR@10 |  MRR |  MAP@3  | MAP@5 | MAP@10  |
> > |------|----------|------|----------|------|----------|----------|----------|
> > |OCEAN + GraphSage		| 1	| 1	| 1 	| 1	| 1	| 1	| 1	 |
> > |OCEAN + GAT		        | 1	| 1	| 1 	| 1	| 1	| 1	| 1	  |
> > |OCEAN + GCN 	                | 0.75	| 1	| 1	| 0.875	| 0.917  | 0.95	| 0.975 |
> >
> > - **Several implementation details are omitted, such as the architecture of the MLP used in L_MI and the hyperparameters.**
> >
> > A: We would like to point out that there is no specific architecture for MLP used in L_MI. There is only the one layer MLP followed by ReLU activation function, such as torch.nn.Linear with ReLU in PyTorch. We provide the hyperparameter analysis in Section 4.2 and Appendix E.3.

---

> > > ### Comment · Reviewer_jmm7 · 2024-11-22
> > >
> > > I appreciate the additional results. However, the authors misunderstood my comment on the comparison with GNN-based causal discovery methods. After NOTEARS, there are GNN-based structure learning methods developed, e.g., DAG-GNN, etc.

---

> > > > ### Author Response · Authors · 2024-11-23
> > > > **Reply by Authors**
> > > >
> > > > Thanks for your clarification. We would like to point out that we have compared a few GNN-based causal discovery methods in our experiment, such as MULAN, REASON, CORAL. These methods are the most recent GNN-based causal discovery methods published in 2023 and 2024. As for DAG-GNN, we find that the encoder of DAG-GNN is MLP. Thus, this method is not designed to deal with the time-series data and experiment on Product Review, Train Ticket and Online Boutique datasets. Please let us know if you have any additional question.

---

> > ### Comment · Reviewer_jmm7 · 2024-11-25
> >
> > Take Figure2b for example, how to choose lambda_2 and lambda_3?

---

### Official Review · Reviewer_j18J · 2024-10-31

**Soundness:** 1
**Presentation:** 3
**Contribution:** 1
**Rating:** 3
**Confidence:** 5

**Summary:**

This paper proposes OCEAN as an online RCA method for multi-modal data in microservice systems. It combines dilated CNNs and GNNs to model temporal dependencies and causal relationships, with a multi-factor attention mechanism and a graph fusion module for cross-modal integration. Experiments show OCEAN’s effectiveness and efficiency in real-time RCA.

**Strengths:**

1. Clear presentation of RCA, especially in the area of multi-modal RCA.
2. Extensive experimental validations.
3. I beileve that the multi-modal information is of importance for RCA, as the combination from diverse sources might be beneficial.

**Weaknesses:**

1. **Main concern 1**:  The necessity of introducing causal discovery (CD) into root-cause analysis.
- RCA is a differenct task from CD,as the former requires identifying a subset of variables while the latter requires the identification of orientations.
- When the prior knowledge is present, e.g., in some cases of microservices, the causal graph is already given. When the prior knowledge is not sufficient, the discovery of causal graph is lack of validation, and the resulting RCA becomes wield.
- My suggestion is that, based on the SCM model, designing some statistics from the data rather the first-discover-then-identify approach, as the latter paradigm is somehow incremental and risky.

2. **Main Concern 2**: As this paper is not the first work to introduce multi-modal information into RCA, I think that this paper should focus on building theoretical understanding on why and how multi-modal information will be beneficial towards RCA. The contribution of this paper, including a causal-structure learning module and a temporal learning framework, seems very incremental such that I do not agree that this paper's novelty reaches the bar of ICLR.

**Questions:**

See Weaknesses

---

> ### Author Response · Authors · 2024-11-22
> **Reply by Authors**
>
> Thank you for your invaluable feedback. We would like to address your primary concerns and provide a response below.
>
> **Main concern 1: The necessity of introducing causal discovery (CD) into root-cause analysis. RCA is a differenct task from CD,as the former requires identifying a subset of variables while the latter requires the identification of orientations. When the prior knowledge is present, e.g., in some cases of microservices, the causal graph is already given. When the prior knowledge is not sufficient, the discovery of causal graph is lack of validation, and the resulting RCA becomes wield. My suggestion is that, based on the (Structural Causal Model) SCM model, designing some statistics from the data rather the first-discover-then-identify approach, as the latter paradigm is somehow incremental and risky.**
>
> A: We agree with your opinion that prior knowledge (e.g., physical dependency graph) provides important information to us. However, we would like to point out that the complete physical dependency graph might not be available in many real-world applications. We experiment on two semi-synthetic datasets (i.e., Train Ticket and Online Boutique datasets) and one real-world dataset, i.e., Product Review dataset. Both Train Ticket and Online Boutique datasets do not include the physical dependency graph. In the Product Review dataset, the physical dependency graph is incomplete and it only consists of around 30 nodes/system entities for each case. Nevertheless, the entire dataset consists of more than 200 nodes/system entities, which indicates that the physical dependency graph can only capture a small percentage of the nodes/system entities and this graph could not be directly used for root cause analysis.
> However, we would like to express our disagreement towards the statement that “When the prior knowledge is not sufficient, the discovery of causal graph is lack of validation, and the resulting RCA becomes wield”. Many existing RCA methods [1,2,3,4] have demonstrated the effectiveness of our proposed method when a physical dependency graph is unavailable.
>
> [1] Dongjie Wang, Zhengzhang Chen, Yanjie Fu, Yanchi Liu, and Haifeng Chen. Incremental causal graph learning for online root cause analysis. In Proceedings of the 29th ACM SIGKDD Conference on Knowledge Discovery and Data Mining, pp. 2269–2278, 2023a.
> [2] Lecheng Zheng, Zhengzhang Chen, Jingrui He, and Haifeng Chen. MULAN: multi-modal causal structure learning and root cause analysis for microservice systems. In Tat-Seng Chua, Chong-Wah Ngo, Ravi Kumar, Hady W. Lauw, and Roy Ka-Wei Lee (eds.), Proceedings of the ACM on Web Conference 2024, WWW 2024, Singapore, May 13-17, 2024, pp. 4107–4116. ACM, 2024
> [3] Dongjie Wang, Zhengzhang Chen, Jingchao Ni, Liang Tong, Zheng Wang, Yanjie Fu, and Haifeng Chen. Interdependent causal networks for root cause localization. In Proceedings of the 29th ACM SIGKDD Conference on Knowledge Discovery and Data Mining, KDD 2023, Long Beach, CA, USA, August 6-10, 2023, pp. 5051–5060. ACM, 2023c
> [4] Roxana Pamfil, Nisara Sriwattanaworachai, Shaan Desai, Philip Pilgerstorfer, Konstantinos Georgatzis, Paul Beaumont, and Bryon Aragam. DYNOTEARS: structure learning from time-series data. In Silvia Chiappa and Roberto Calandra (eds.), The 23rd International Conference on Artificial Intelligence and Statistics, AISTATS 2020, 26-28 August 2020, Online [Palermo, Sicily, Italy], volume 108 of Proceedings of Machine Learning Research, pp. 1595–1605. PMLR, 2020
>
> **Main Concern 2: As this paper is not the first work to introduce multi-modal information into RCA, I think that this paper should focus on building theoretical understanding on why and how multi-modal information will be beneficial towards RCA. The contribution of this paper, including a causal-structure learning module and a temporal learning framework, seems very incremental such that I do not agree that this paper's novelty reaches the bar of ICLR.**
>
> A: We would like to point out that our work is NOT solely targeting multi-modal root cause analysis. Our method is an **ONLINE multi-modal root cause method**, and the existing methods are **OFFLINE** multi-modal root cause methods. Each component in our proposed method is designed for the online root cause analysis setting and ignoring this specific setting is unfair to the evaluation of our contribution.

---

> ### Comment · Reviewer_j18J · 2024-11-24
> **Response**
>
> By the way, existing RCA methods on online data also exist. So my critical question stills exists, i.e., what is the unique benefit (either theoretical or empirical results) when introducing multi-modal setting?

---

### Official Review · Reviewer_xQ4N · 2024-11-02

**Soundness:** 3
**Presentation:** 2
**Contribution:** 3
**Rating:** 5
**Confidence:** 3

**Summary:**

This article proposes the OCEAN model for anomaly detection in microservice systems. It uses diffuse convolution to capture long-term temporal dependencies, introduces two modalities of data, log data and metric data, and adaptively models the causal relationship structure between and within data based on the attention mechanism. It also develops a contrast graph fusion module based on mutual information maximization to effectively model the relationship between various modalities.

**Strengths:**

Quality: The experimental workload of the article is relatively substantial.
Significance: It provides a multi-modal solution for anomaly detection in online micro-systems.

**Weaknesses:**

-This paper introduces diffuse convolution to model long-term temporal dependencies, attention mechanism to model causal structures, and mutual information fusion causal graphs, and describes and experiments with them as the core innovations of this paper. However, these three methods have been around for a long time, and the author did not explain well how this paper improves them.
-Causal graph learning is an important component of the OCEAN model, but lack of corresponding causal analysis or experimental results.
-There are many errors in the article's consistency of symbols and textual expression.

**Questions:**

1. On page 2, line 83, the author introduces a factor attention mechanism to analyze the relationship between different factors and re-evaluate their impact on online causal graph learning. As far as I know, many methods use attention mechanisms to model and analyze the relationship between different variables, as shown in references 1-2. What is the essential innovation of this paper compared with them?
[1] Wu X, Ajorlou A, Wu Z, et al. Demystifying over-smoothing in attention-based graph neural networks[C]. Advances in NeurIPS, 2024.
[2] Cai J, Zhang M, Yang H, et al. A novel graph-attention based multimodal fusion network for joint classification of hyperspectral image and LiDAR data[J]. Expert Systems with Applications, 2024, 249: 123587.
2. Page 4, line 186 indicates that this paper uses the 2021 MSSA algorithm and claims it is the most advanced online fault detection method. Why is the most advanced online fault diagnosis method from 2021? We searched for several online fault diagnosis algorithms published in recent years as shown in references 1-2.
[1] Zeiser A, Özcan B, van Stein B, et al. Evaluation of deep unsupervised anomaly detection methods with a data-centric approach for on-line inspection[J]. Computers in Industry, 2023, 146: 103852.
[2] Wang X, Yao Z, Papaefthymiou M. A real-time electrical load forecasting and unsupervised anomaly detection framework[J]. Applied Energy, 2023, 330: 120279.
3. The dimensions of \textbf{\emph{H}}_{0}^{M}[\emph{j}]^{T} and \textbf{\emph{W}}^{3} in Formula 9 are [T_3 \times \emph{d}_{M}] and [T_3 \times T_3] respectively. Why can they be directly matrix multiplied?
4. Among the 7 compared algorithms, is it fair to compare them with 4 algorithms that focus on learning causal graphs rather than fault diagnosis algorithms? We list some recent fault diagnosis algorithms as shown in the references 1-3.
[1] Chen, D., Liu, R., Hu, Q., & Ding, S. X.. Interaction-aware graph neural networks for fault diagnosis of complex industrial processes. IEEE Transactions on neural networks and learning systems, 2021, 34(9), 6015-6028.
[2] Liu Y, Jafarpour B. Graph attention network with Granger causality map for fault detection and root cause diagnosis[J]. Computers & Chemical Engineering, 2024, 180: 108453.
[3] Zhou Q, Pang G, Tian Y, et al. AnomalyCLIP: Object-agnostic Prompt Learning for Zero-shot Anomaly Detection[C]. The Twelfth International Conference on Learning Representations,2024.
5. The authors show that the proposed method can learn inter-modal and intra-modal causal graphs. Can the learned causal graph structure be further demonstrated experimentally?
6. There are many errors in the organization logic, symbol unification and text expression of the article, so I suggest careful revision. For example: \emph{d}_{M} in line 158 does not match the description given in Table 1; two identical \textbf{\emph{a}}^{0}_{L}[\emph{j}] appear in line 291 of page 6; there is missing punctuation before “so that” in line 304, etc.

---

> ### Author Response · Authors · 2024-11-22
> **Reply by Authors**
>
> Thank you for your invaluable feedback. We would like to address your primary concerns and provide a response below.
>
> **Q1. On page 2, line 83, the author introduces a factor attention mechanism to analyze the relationship between different factors and re-evaluate their impact on online causal graph learning. As far as I know, many methods use attention mechanisms to model and analyze the relationship between different variables, as shown in references 1-2. What is the essential innovation of this paper compared with them? [1] Wu X, Ajorlou A, Wu Z, et al. Demystifying over-smoothing in attention-based graph neural networks[C]. Advances in NeurIPS, 2024. [2] Cai J, Zhang M, Yang H, et al. A novel graph-attention based multimodal fusion network for joint classification of hyperspectral image and LiDAR data[J]. Expert Systems with Applications, 2024, 249: 123587.**
>
> A: We would like to point out that the attention mechanism used in [1,2] and the attention mechanism in our method are quite different. (1). In [1,2], the attention mechanism aims to measure the importance of the neighbors around the anchor node, while our proposed method aims to capture the interrelationship between two modalities and meanwhile measure the importance of each factor. Basically, the attention map used in [1,2] measures the importance of one node to another node, while the attention map used in our method measures the inter-connection between two factors from two modalities.
>
> **Q2. Page 4, line 186 indicates that this paper uses the 2021 MSSA algorithm and claims it is the most advanced online fault detection method. Why is the most advanced online fault diagnosis method from 2021? We searched for several online fault diagnosis algorithms published in recent years as shown in references 1-2. [1] Zeiser A, Özcan B, van Stein B, et al. Evaluation of deep unsupervised anomaly detection methods with a data-centric approach for on-line inspection[J]. Computers in Industry, 2023, 146: 103852. [2] Wang X, Yao Z, Papaefthymiou M. A real-time electrical load forecasting and unsupervised anomaly detection framework[J]. Applied Energy, 2023, 330: 120279.**
>
> A: We appreciate your suggestions. We would like to point out that the proposed method in [2] is “a combination of WGAN and encoder CNN, adapted from f-AnoGAN (published in 2019)”, as stated in the abstract of [1]. In addition, [1] is designed to detect anomalies for image data rather than time-series. For [2], although [2] is designed to detect the anomalies for time-series data in the unsupervised setting, it requires the model trained on the time-series data first, then forecast the future values and finally detect the anomaly based on whether the change is within a reasonable range. According to [2], in the experiment, it usually requires the length of the data to be more than 1 year as shown in section 5.1.2 for data preprocessing. However, in our experiment, the largest dataset, (i.e., Product Review Dataset) only consist of 2 day time series data, not to mention two small datasets. This suggests that using [2] to detect anomalies might not be a good choice. Here, we would like to emphasize that the main reason why we select 2021 MSSA algorithm to detection the change point is that this paper has **a good performance with solid theoretical performance guarantee**.
>
> **Q3 & Q6. There are many errors in the organization logic, symbol unification and text expression of the article, so I suggest careful revision.**
>
> A: We have corrected all of these typos in our updated version and highlighted the change in red.
>
> **Q4. Among the 7 compared algorithms, is it fair to compare them with 4 algorithms that focus on learning causal graphs rather than fault diagnosis algorithms? We list some recent fault diagnosis algorithms as shown in the references 1-3.**
>
> A: We would like to point out that the problem setting in our paper is different from the setting in papers [1] and [3].
> - Paper [1] assumes that the label information of system faults in the training set are available and system faults in both the training set and the test set are overlapped. With this assumption, [1] aims to learn a classifier trained on the labeled samples from the training set and then predicts the unlabeled samples in the test set. However, in our setting, we do not have such an assumption and no labeled samples/system faults are available to train such a classifier.
> - Paper [3] is designed for anomaly detection on the object and it is not designed for root cause identification in a microservice system. These two are totally different settings.
> - Paper [2] shares a similar experimental setting with our paper. However, we failed to find the source code either in the GitHub or the author’s homepage.

---

> > ### Author Response · Authors · 2024-11-22
> > **Reply by Authors**
> >
> > **Q5. The authors show that the proposed method can learn inter-modal and intra-modal causal graphs. Can the learned causal graph structure be further demonstrated experimentally?**
> >
> > A:Here, we evaluate the quality of the learned causal graph by comparing it with the physical dependency graph with two settings. In the first setting, we compared the causal graph learned by each modality (corresponding to the inter-modal graphs) and in the second setting, we compared the fused causal graph from two modality (corresponding to the intra-model graph).
> > Following Dynotear [1], we use AUROC and SHD as two metrics to quantify the difference between learned causal graphs and the physical dependency graph.
> >
> > |Graphs | SHD | AUROC |
> > |------|------|------|
> > |Metric modality        | 0.314 | 0.865 |
> > |Log modality            | 0.593 | 0.663 |
> > |Fused causal graph | 0.298 | 0.881|
> >
> > [1] Pamfil, Roxana, Nisara Sriwattanaworachai, Shaan Desai, Philip Pilgerstorfer, Konstantinos Georgatzis, Paul Beaumont, and Bryon Aragam. "Dynotears: Structure learning from time-series data." In International Conference on Artificial Intelligence and Statistics, pp. 1595-1605. Pmlr, 2020.

---

### Official Review · Reviewer_eBR8 · 2024-11-02

**Soundness:** 2
**Presentation:** 2
**Contribution:** 2
**Rating:** 3
**Confidence:** 4

**Summary:**

In this paper, the authors proposed a new online causal structure learning method from time series which is then evaluated in microservice systems RCA problem. The authors combine a so-called dilated CNN and GCN to learn the structure by autoregressively forecasting the future time series. In addition, an attention mechanism is used to reweight and fuse the learned graph. Finally, random walk with restart is used to determine the root cause using the learned graph. Experiments show the effectiveness of the proposed method.

**Strengths:**

1. The proposed method seems to be effective, which outperforms many existing methods.

**Weaknesses:**

1. The introduction and problem definition is not clear, especially the multi-model part and online setting part. The paper is more about online causal structure learning from time series and is evaluated in the microservice setting. The authors are suggested to change the title and rewrite the introduction to stand out the focus of the paper.
2. The experimental setting is not clear. The online evaluation setting is not described, nor the online problem setting. The authors are suggested to define the online problem setting and clearly describe the evaluation setting.
3. The source code is not provided, which makes reproducibility of the paper low. The authors are suggested to open source the code.

**Questions:**

1. Please clearly define online / offline RCA in introduction. RCA is usually triggered by KPI anomalies. Therefore, it is not a function that needs to be conducted continuously, like anomaly detection. I believe many of the existing works have been deployed in the production system. Are they online RCA methods according to the authors' definition? It is not clear to me that the authors stated that most existing methods are designed for offline use. Methods can be trained offline and used online.
2. The example in the introduction stating that log is necessary apart from metrics is not convincing. "Disk Space Full" can be solely identified by metrics. Regarding "Database Query Failures", what specific problem do the authors want to identify? For which kind of system OLTP or OLAP? Why is solely using metrics not enough?
3. When talking about multi-modal data, the authors metrics and logs, why not consider trace, which seems to be more important for microservice systems. For instance, [1] proposed a method deployed in the production system which considers both metrics and traces and this work is overlooked by the authors.
4. "T1", "T2", "n-1" and $d_M$ are defined without usage in line 156.
5. It seems that the authors converted log to metric data. The problem is only defined on metric data. What is the difference between the old metrics and new metrics converted from log? For me, at least the method is only for single modal data. It is not clear why existing single model methods cannot be applied in such a setting.
6. In the problem definition, the authors ignore the physical relation but propose to use a purely data-driven model to learn the causal structure. Relationships like which service calls the other service, which pod is in which virtual machines and which physical machines, are known but ignored. Can the authors explain the reason behind such a choice? Which practical scenario matches the setting that the authors would like to study?
7. After reading Section 3, I found that the main focus of the paper is online causal graph learning from time series data. The title and introduction is way broader than the studied problem, which does not match to the content of the paper from my point of view. Moreover, in the microservice setting, why would the causal graph be changing overtime if it is a causal graph reflecting the ground truth? The authors are suggested to better motivate this point.
8. The proposed model seems to be very complex with several components. How many weights does the model own? How do the authors avoid overfitting and catastrophical forgetting problems during online structure learning?
9. In Table 2, the proposed method can be used for both metric only and log only settings. The authors are suggested to give both results.
10. The experimental setting is not clear. Since the proposed method is online, so when will the proposed method be used? What is the batch used for evaluation? And after how batches will the method be evaluated? Again, this confusion may be due to the fact that the online setting is not clearly defined.
11. It seems that there is no code to reproduce the experiments. In addition, did the authors reproduce the results from other methods or copy the number from their paper? Please give the specific parameter setting or state clearly that the numbers are copied from the papers.

[1] ShapleyIQ: Influence Quantification by Shapley Values for Performance Debugging of Microservices. ASPLOS 2024.

---

> ### Author Response · Authors · 2024-11-22
> **Reply by Authors**
>
> Thank you for your invaluable feedback. First, we would like to identify a few factuality misunderstood by Reviewer 2.
>
> **Q1: Please clearly define online / offline RCA in introduction. RCA is usually triggered by KPI anomalies. Therefore, it is not a function that needs to be conducted continuously, like anomaly detection. I believe many of the existing works have been deployed in the production system. Are they online RCA methods according to the authors' definition? It is not clear to me that the authors stated that most existing methods are designed for offline use. Methods can be trained offline and used online.**
>
> A: We would like to point out the main differences of definition between online RCA and offline RCA as follows:
> - In the offline setting, only the historical data is available for model training, but the batch data is available and serves as very important indicators for the root cause analysis in the online setting.
> - **There is no good way to leverage the batch data for the offline methods.** In the online setting, our proposed method will first be initialized by training the model with the historical data only and it is only fine-tuned for around 100 iterations for each batch data. In the offline setting, the model needs to be trained from scratch when new data is provided, which can be time-consuming. In addition, we also examine the performance of the model fine-tuned on newly available batch data. The experimental results show that fine-tuning these multi-modal RCA models yields even worse performance than training the model from scratch. This is the major reason why an online RCA method is necessary compared with the offline RCA methods.
>
> **Training from scratch (Two Modalities)**
> | Model | PR@1 | PR@5 | PR@10 | MRR | MAP@3 | MAP@5 | MAP@10|
> |------|----------|------|----------|-----|----------|------|----------|
> |Dynotears 	| 0       | 0.25  | 0.50 | 0.092  | 0        | 0.05 | 0.175 |
> |C-LSTM 	| 0.25  |0.5     | 0.5   | 0.409  | 0.417 | 0.45 | 0.475 |
> |GOLEM 	| 0       |0        | 0.25 | 0.043  | 0        | 0      | 0.025 |
> |REASON 	| 0.25  |1.0     | 1.0   | 0.562  | 0.583 | 0.75 | 0.875 |
> | MULAN 	| 0.75 | 1.0    |  1.0   | 0.833 | 0.833 | 0.9    | 0.95 |
>
> **Fine-tuning the model  (Two Modalities)**
> |Model | PR@1 | PR@5 | PR@10 | MRR | MAP@3 | MAP@5 | MAP@10 | Average Performance Drop|
> |------|----------|------|----------|-----|----------|------|----------|------|
> |Dynotears 	| 0       | 0        | 0.50 | 0.093  | 0        | 0.0    | 0.125 | -0.06 |
> |C-LSTM 	| 0       |0.25    | 0.5   | 0.184  | 0.167 | 0.2    | 0.325 | -0.141 |
> |GOLEM 	| 0       |0         | 0      | 0.056  | 0        | 0       | 0.0     | -0.044 |
> |REASON 	| 0.25  |1.0      | 1.0   | 0.521  | 0.5     | 0.70  | 0.85   | -0.105 |
> |MULAN		| 0.25  | 1.0     | 1.0   | 0.583  | 0.667 | 0.8    | 0.9     | -0.152|
>
> We highlight the difference between online setting and offline setting in section 3.1 in red in the updated version.
>
>
> **Q5: It seems that the authors converted log to metric data. The problem is only defined on metric data. What is the difference between the old metrics and new metrics converted from log? For me, at least the method is only for single modal data. It is not clear why existing single model methods cannot be applied in such a setting.**
>
> A: We would like to point out that you misunderstand that. We **DO NOT** convert log data to metric data. Instead, we only convert log into time-series data described in Line 158-160, which has the similar format as metric data. We also provide the details of log feature extraction in Appendix C. Therefore, the log time-series data **is different** from metric data.
>
> **Q9. In Table 2, the proposed method can be used for both metric only and log only settings. The authors are suggested to give both results.**
>
> A: We want to point out that you misunderstand our proposed method. First, log data are different from metric data and we do not convert log data to metric data. Second, our proposed representation learning with multi-factor attention aims to capture the interaction between metric and log data. Our proposed method **CANNOT** only experiment on a single data type.

---

> > ### Author Response · Authors · 2024-11-22
> > **Reply by Authors**
> >
> > Next, we would like to address your primary concerns and provide a response below.
> >
> > **Q2: The example in the introduction stating that log is necessary apart from metrics is not convincing....**
> >
> > A: We agree with you that “Disk Space Full can be solely identified by metrics”, but our statement is that “issues like “Disk Space Full” are more effectively identified through combined analysis of metrics and logs”. We aim to show that adding logs data can identify the root cause in a more effective way as log data might convey some extra information. For "Database Query Failures", we do not want to constrain our method in the application of OLTP or OLAP. To detect the system fault like “Database Query Failures”, log data contains more evident information than system metrics, such as error messages like "query execution failed," "connection timeout," or "syntax error.". However, metric data lacks the rich context provided by logs, such as the exact query, parameters, and specific error messages. This makes it difficult to pinpoint issues related to specific queries or edge cases if only metrics are available.
> >
> > **Q3.When talking about multi-modal data, the authors metrics and logs, why not consider trace, which seems to be more important for microservice systems. For instance, [1] proposed a method deployed in the production system which considers both metrics and traces and this work is overlooked by the authors. [1] ShapleyIQ: Influence Quantification by Shapley Values for Performance Debugging of Microservices. ASPLOS 2024.**
> >
> > A: We would like to point out that OCEAN can naturally extend to include additional modalities, such as traces. These types of data can enhance the model's ability to capture complex interactions and dependencies within the system. We conducted additional experiments by incorporating traces into the AIOps and Train Ticket datasets. The results demonstrated improved performance, as the inclusion of traces provided valuable context and enriched the causal structure learning. This additional information allows OCEAN to more accurately identify root causes and improve the precision of its analysis.
> >
> > |Product Review	| PR@1 |  PR@5  |  PR@10 |  MRR |  MAP@3  | MAP@5 | MAP@10  | Time (s) |
> > |------|----------|------|----------|------|----------|------|----------|----------|
> > |OCEAN			| 1	| 1	| 1 	| 1	| 1	| 1	| 1	|  20.1 |
> > |OCEAN + trace		| 1	| 1	| 1	| 1	| 1	| 1	| 1	|  26.3 |
> > |Train Ticket | PR@1 |  PR@5  |  PR@10 |  MRR |  MAP@3  | MAP@5 | MAP@10  | Time (s) |
> > |OCEAN			| 0.2	| 0.4	| 1	| 0.38	| 0.33	| 0.36	| 0.58	|  3.2 |
> > |OCEAN + trace 		| 0.2	| 0.6	| 1	| 0.39	| 0.33	| 0.44	| 0.62	|  3.8 |
> >
> > **Q4."T1", "T2", "n-1" and d_M are defined without usage in line 156.**
> >
> > A: We would like to clarify that we want to explicitly provide each notation for the symbol used in our paper, even if T1 and T2 are only used to describe the length of historical metric data and the length of each batch. For n-1, we aim to specify the number of system entities is equal to n-1 and n is used in many places, such as the feature dimension of W^i in equation (1), equation 8. Similarly, d_M is also used in many equations, such as equation 1, equation 13, equation 16, etc.
> >
> >
> > **Q6. In the problem definition, the authors ignore the physical relation but propose to use a purely data-driven model to learn the causal structure....**
> >
> > A: We appreciate your suggestions of incorporating the physical dependency graph into the proposed method. We agree that $A_{old}$ in our method can be initialized based on the prior knowledge of physical dependency graph. Here, we evaluate the quality of the learned causal graph by comparing it with the physical dependency graph with two settings. In the first setting, we compared the causal graph learned by each modality (corresponding to the inter-modal graphs) and in the second setting, we compared the fused causal graph from two modality (corresponding to the intra-model graph).
> > Following Dynotear [1], we use AUROC and SHD as two metrics to quantify the difference between learned causal graphs and the physical dependency graph.
> >
> > |Graphs | SHD | AUROC |
> > |------|------|------|
> > |Metric modality        | 0.314 | 0.865 |
> > |Log modality            | 0.593 | 0.663 |
> > |Fused causal graph | 0.298 | 0.881|
> >
> > [1] Pamfil, Roxana, Nisara Sriwattanaworachai, Shaan Desai, Philip Pilgerstorfer, Konstantinos Georgatzis, Paul Beaumont, and Bryon Aragam. "Dynotears: Structure learning from time-series data." In International Conference on Artificial Intelligence and Statistics, pp. 1595-1605. Pmlr, 2020.

---

> > > ### Author Response · Authors · 2024-11-22
> > > **Reply by Authors**
> > >
> > > **Q7. After reading Section 3, I found that the main focus of the paper is online causal graph learning from time series data...**
> > >
> > > A: We partially agree with your statement that most part of casual graph (denoted as $A_{old}$ in our method) should remain unchanged, but we would like to argue that the rest part of the causal graph (denoted as $\Delta A_v$ in our method) might change from time to time due to the emergence of the system fault. Thus, in our paper, we partition the causal graph into two parts, i.e.,   $A_{old}$ and $\Delta A_v$. We plan to change the title to “Online Multi-modal Root Cause Analysis for time-series data in Microservice Systems”.
> > >
> > > **Q8. The proposed model seems to be very complex with several components. How many weights does the model own?**
> > >
> > > A: The number of weights depends on the size of the dataset. Take the largest dataset (Product Review dataset) for instance, one system fault consists of 207 system entities, 600 timestamps for the historical data and 24 timestamps for the batch data (165 batches). The size of the input data is 207x(24x165+600)x(5+2)=6,607,440. The total number of the parameters in our model is 557,099.
> > >
> > > **Q10. The experimental setting is not clear. Since the proposed method is online, so when will the proposed method be used? What is the batch used for evaluation? And after how batches will the method be evaluated? Again, this confusion may be due to the fact that the online setting is not clearly defined.**
> > >
> > > A: In the online setting, every several minutes, a new batch data (e.g., i-th batch) is given. Once the online failure detection is triggered, starting from the i-th batch, we keep finetuning the model with equation 18. Our proposed method is not only fine tuned with the i-th batch but also the next several batches, which aims to ensure the correctness of the final results. We provide the stopping criteria in Section 3.5.
> > >
> > > **Q11. It seems that there is no code to reproduce the experiments. In addition, did the authors reproduce the results from other methods or copy the number from their paper? Please give the specific parameter setting or state clearly that the numbers are copied from the papers.**
> > >
> > > A: We will release our code when our paper is accepted. We reproduce the results with the default parameter setting in their source codes, which is available in a public GitHub link (https://github.com/lemma-rca/rca_baselines/tree/main).

---

> > > > ### Comment · Reviewer_eBR8 · 2024-11-25
> > > > **Thanks for the response.**
> > > >
> > > > Thanks for the response. However, the formal definition of online RCA is still missing. As I wrote before, I still think the studied problem is online causal graph learning. I am appreciated that the authors conduct new experiments using the tracing data. However, how are they used is not clear as well. Moreover, sorry, I did not get the point of new experiments regarding learned casual graph and physical graph. Anyway, I think the paper needs significant revision with clear problem definition, motivation for learning casual graph, using trace data (or not), using physical graph (or not) and corresponding experimental design. At current stage, it is not ready for publication. Therefore, I will retain my evaluation.

---

### Official Review · Reviewer_YMLu · 2024-11-04

**Soundness:** 3
**Presentation:** 3
**Contribution:** 2
**Rating:** 5
**Confidence:** 4

**Summary:**

The paper introduces the OCEAN framework for online multi-modal root cause analysis (RCA) in microservice systems, showing significant improvements in accuracy and computational efficiency across multiple real-world datasets. However, it is critiqued for its limited methodological innovation. While the approach achieves good results, it primarily builds on existing techniques, such as dilated convolution, which have been extensively studied in related research. The paper focuses on addressing common challenges in multi-modal learning that have already been explored, rather than tackling the specific and unique challenges of multi-modal RCA. Additionally, it lacks a clear explanation of how its methods enhance real-time processing and reduce resource consumption. Minor issues include inconsistencies in the font of an equation.

**Strengths:**

This paper presents a novel and highly effective approach to online multi-modal root cause analysis in microservice systems. The proposed OCEAN framework showcases impressive advancements in both accuracy and computational efficiency, achieving good results across multiple real-world datasets.

**Weaknesses:**

Limited Innovation and Contribution: While the paper achieves good results, it lacks methodological innovation specifically for online root cause analysis. For example, the core method used to reduce computational time, dilated convolution, has been previously employed. Although the appendix discusses its time complexity in comparison to LSTM and Transformers, paper [1] as early as 2018 introduced the use of dilated convolutional neural networks for capturing temporal dependencies. Furthermore, paper [2] in 2022 utilized this approach specifically for capturing long-term temporal dependencies.
The proposed methods primarily address common issues (C1,C2, and C3) in multi-modal learning and do not make substantial progress in tackling the unique challenges of root cause analysis in the multi-modal domain. For example, Method 2, as referenced in the paper, utilizes approaches similar to those in MULAN[3] to extract multi-modal features from offline datasets. However, this paper lacks a detailed explanation of the potential relationships among factors from both modalities and does not clarify why it achieves better results than MULAN. Furthermore, in Method 3, Learning Multi-modal Causal Structures, the paper does not adequately address modality reliability or the reliability of the causal graph, leaving the approach lacking in clear interpretability.
Motivation Not Clear: The primary novelty of this paper appears to focus on "online" multi-modal root cause analysis (RCA) for microservice systems. However, the methodology introduced lacks a clear explanation of how it enhances real-time processing capabilities or reduces resource consumption in practice. The paper proposes the use of dilated convolutional neural networks (DCNNs) and a graph-based approach but does not sufficiently justify how these choices directly contribute to improved real-time performance or lower computational overhead.
•Minor Issues: The font in Equation (13) is somewhat inconsistent.
[1] Borovykh, Anastasia, Sander Bohte, and Cornelis W. Oosterlee. "Dilated convolutional neural networks for time series forecasting." Journal of Computational Finance, Forthcoming (2018). [2] Ayodeji A, Wang Z, Wang W, et al. Causal augmented ConvNet: A temporal memory dilated convolution model for long-sequence time series prediction[J]. ISA transactions, 2022, 123: 200-217 [3] Zheng, L., Chen, Z., He, J., & Chen, H. (2024, May). MULAN: Multi-modal Causal Structure Learning and Root Cause Analysis for Microservice Systems. In Proceedings of the ACM on Web Conference 2024 (pp. 4107-4116).

**Questions:**

In the "weaknesses" part, it is essential to address all the issues mentioned, particularly the concern that the proposed methods appear to be direct employment of existing approaches and seem to target general problems in multimodal scenarios rather than being specifically tailored for multimodal Root Cause Analysis (RCA). Additionally, a clear explanation is needed as to why the proposed methods can improve real-time performance and reduce resource consumption. This should be supported by supplementary experimental evidence.

---

> ### Author Response · Authors · 2024-11-22
> **Reply by Authors**
>
> Thank you for your invaluable feedback. We would like to address your primary concerns and provide a response below.
>
> W1. **Limited Innovation and Contribution: While the paper achieves good results, it lacks methodological innovation specifically for online root cause analysis.....**
>
> A: We would like to differentiate the difference between our proposed method and these papers.
>
> - **Difference between our method and [1] and [2]:** Indeed, papers [1] and [2] can capture the long-term temporal dependencies. Our contribution in Section 3.2 is to capture both temporal dependency and the causal relationship by minimizing the forecasting errors in equation 8. This is different from [1] and [2]. Notice that [2] could not capture the causal relation between two system entities (e.g., system entity X -> system entity Y), which is our major goal in Section 3.2.
>
> - **Difference between our method and Mulan [3]:** Though both MULAN and our method uses contrastive learning loss to maximize the similarity between two modalities, our method also aims to capture the interrelationship among two modalities in section 3.3. This section aims to measure how different factors within two modalities correlate to each other and how these factors contribute to causal structure learning, which is ignored by Mulan. **In line 276-282, we explicitly introduce the way to capture the relationship among different factors from both modalities via equation 9.** We contribute the superior performance of our method over Mulan to the design of representation learning with multi-factor attention in section 3.3.
>
> - **How do method 3 address modality reliability:** We would like to point out that the third change (C3) Learning Multi-modal Causal Structures is addressed by the combination of section 3.2 and 3.3. In Section 3.2, we introduce **the similarity matrix $\textbf{C}$ to measure how different factors between two modalities contribute to causal structure learning** via equation 9. Specifically, the smaller value in the similarity matrix $\textbf{C}$ indicates that this factor contributes less to the causal structure learning and a large value indicates more contribution this factor made towards causal structure learning. Based on this design, we **assess the reliability of each modality based on the importance of factors within each modality via the importance measurement** in equation 16. Typically, a larger value of $s_M$ implies that the metric data is more important than the log data. To better assess the reliability of each model by measuring the weight for each modality, we conduct a case study below. In the table, we observe that our proposed method assigns larger weights to metric data across four cases, which is consistent with the observation in the experimental results in Table 2 that all baseline methods achieve better performance with only metric data than those with the log data.
>
> | Modality	| Case 1 | Case 2 | Case 3 | Case 4|
> | -----|-------|--------|--------|--------|
> | Metric weight 	| 0.673  | 0.526  | 0.764  | 0.687 |
> | Log weight	        | 0.327  | 0.474  | 0.236  | 0.313 |
>
>
> W2. **Motivation Not Clear: The primary novelty of this paper appears to focus on "online" multi-modal root cause analysis (RCA) for microservice systems.**
>
> A:  Here is our special design for the online multi-modal RCA. In the preparation (offline) phase, we will first train the model with only the historical data by minimizing equation 18. Notice that in the preparation phase, the batch data is not available and thus we remove all loss terms involving the batch data. In the online phase,  batch data is available and we only need to fine-tune the model for 100 iterations by minimizing equation 18.
>
> W3. **Minor Issues: The font in Equation (13) is somewhat inconsistent.**
>
> A: We have updated the font in equation 13.

---

### Note · Authors · 2024-12-01

I have read and agree with the venue's withdrawal policy on behalf of myself and my co-authors.